# M-PESA and Financial Inclusion in Kenya: Of Paying Comes Saving?

**Leo Van Hove** [1,2,*]  **and Antoine Dubus** [3]

1    Department of Applied Economics (APEC), Vrije Universiteit Brussel, Pleinlaan 2, 1050 Brussels, Belgium
2    Faculty of Economic Sciences and Management, Nicolaus Copernicus University, Gagarina 13a,
     87-100 Toruń, Poland
3    Télécom ParisTech, 46 rue Barrault, 75013 Paris, France; antoine.dubus@telecom-paristech.fr
*    Correspondence: Leo.Van.Hove@vub.be; Tel.: +32-2-6292125

**Abstract:** Mobile financial services such as M-PESA in Kenya are said to promote inclusion. Yet only 7.6 per cent of the Kenyans in the 2013 Financial Inclusion Insights dataset have ever used an M-PESA account to save for a future purchase. This paper uses a novel, three-step probit analysis to identify the socio-demographic characteristics of, successively, respondents who do not have access to a SIM card, have access to a SIM but do not have an M-PESA account, and, finally, have an account but do not save on it. We find that those who are excluded in the early stages are predominantly poor, non-educated, and female. For the final stage, we find that those who are in a position to save on their phone—the phone owners, the better educated—are less likely to do so. These results go against the traditional optimistic discourse on mobile savings as a prime path to financial inclusion. As such, our findings corroborate qualitative research that indicates that Kenyans have other needs, and want their money to circulate and 'work'.

**Keywords:** financial inclusion; saving; mobile financial services; M-PESA; Kenya

## 1. Introduction

Broadly defined, financial inclusion refers to "access to and usage of appropriate, affordable, and accessible financial services" [1] (p. 6). Like many other developing countries, Kenya has a low rate of financial inclusion. In 2006—prior to the launch of M-PESA, the leading local mobile money transfer service—only 18.5 per cent of Kenyans used formal services (i.e., mostly bank accounts), 8.1 per cent used semi-formal services (such as those provided by microfinance institutions), 35.0 per cent used the informal sector (rotating savings and credit associations, etc.), and no less than 38.3 per cent were completely excluded; see [2] (Table 1, p. 481) for precise definitions. Even in 2017 only 55.7 per cent of Kenyans over 15 had an account at a formal financial institution, compared to 93.1 per cent in the US [3].

Although informal financial systems are relatively well developed in Kenya [4], it is commonly believed that access to formal financial tools reduces poverty, stimulates investment, and creates growth, particularly in rural areas [5–7]. In particular, the ability to save securely would make a difference, as it increases the resilience of the poor to income shocks and may, ultimately, even enable them to invest in a business. Enabling the rural poor to overcome poverty might also stem migration to cities such as Nairobi, which suffer from overpopulation [8]. Given that financial inclusion may thus contribute to several of the United Nations' Sustainable Development Goals [9]—reducing hunger and poverty, making cities sustainable, and even improving gender equality (see 3.3)—it is not surprising that the promotion of formal saving in developing countries is increasingly attracting the attention of academics. This paper examines the uptake of mobile financial services (MFS) in Kenya. MFS are said

to promote financial inclusion, not only by giving users access to financial services by way of their mobile phone account, but also by supposedly functioning as a stepping stone toward the adoption of a traditional bank account [10] (p. 2).

Kenya is a country where MFS could have a particularly high impact, for a number of reasons. On the demand side, there is the high level of poverty in combination with the importance of rural areas. According to the World Development Indicators of the World Bank [11], in 2015 36.1 per cent of the Kenyan population was poor. The United Nations Food and Agriculture Organization (FAO) estimates that 80 per cent of the population, especially those in rural areas, derive their livelihoods mainly from agricultural related activities [12]. This high dependence on agriculture is a vector of financial instability among rural households, which, in turn, implies that emergency remittances—often from relatives who live in the city—are of vital importance. On the supply side, as mentioned the penetration of bank accounts is low, particularly in rural areas, but adoption of M-PESA is widespread. Kenya is even often presented as "the poster child that 'digital payments can make the world a better place" [13]; see also [14] (p. 412).

In our analysis, we look into which socio-demographic factors promote or inhibit saving on MFS in Kenya. In doing so, we assimilate MFS with M-PESA, representing as it did 96 per cent of all MFS accounts in the country in 2013 [15]. Unlike existing studies, which simply compare MFS users and non-users, we tackle the question in three steps. First, we examine which Kenyans own a SIM card, as this is a precondition for being able to open an M-PESA account. In a second step we then examine which individuals in this subsample have an M-PESA account. In the final step we examine which M-PESA users actually save on their mobile phone account. In each of these steps, we perform our analyses for three, increasingly narrow (sub)samples; the idea being to gradually zoom in on the more vulnerable groups in the population. Tellingly, as we do so, all three indicators that we focus on—ownership of a SIM, possession of an M-PESA account, and saving on M-PESA—go down.

Specifically, we perform probit analysis on data taken from the Financial Inclusion Insights (FII) Program survey conducted by InterMedia in 2013 among 3000 respondents [15] and, in a robustness check, also on data from the FinAccess survey for the same year [16]. We find that those who are 'left behind' in the early steps are predominantly those who—according to the mainstream literature—would benefit most from formal saving, namely the poor, the non-educated, and, indirectly, also women. Moreover, the problem is, by and large, bigger for the rural than for the urban population. We also find that those who are in a position to save on MFS—the better educated, the phone owners—are *less* likely to do so.

Our paper contributes to the literature in several respects. In terms of data, we are the first to use the FII dataset—at least for our purposes. In terms of methodology, compared to the extant studies—which examine only one of the three steps and/or lump together some or all of the steps—our three-step approach allows us to map the relative importance of the different hurdles and identify more accurately the determinants of the 'attrition' at each of the stages. In terms of findings, we bring a sobering note to the more optimistic results of, in particular, Suri and Jack [17], who estimate that M-PESA has lifted 2 per cent of Kenyan households out of poverty. The impact of M-PESA is undoubtedly positive in certain respects, but our results show that it has failed to live up to the high expectations of the mainstream literature when it comes to saving. Traditional savings triggered by MFS would not seem to be the prime path to financial inclusion in Kenya. We argue that MFS should be better tailored to the needs of the Kenyan (rural) population and should take into account the reciprocal and often collective nature of current saving practices.

The remainder of the paper is structured in five sections. Section 2 explains the concept of M-PESA, as well as its state of affairs around the time of the data collection. Section 3 reviews the literature on M-PESA. In Sections 4 and 5 we present, respectively, the survey data and our methodology. Section 6 presents our results and Section 7 concludes.

## 2. M-PESA Identikit

The concept of MFS is simple: digital value is transferred by way of text messages, and a network of agents allows users to withdraw money from or deposit money into their mobile account. In Kenya, M-PESA was launched in 2007 after Safaricom, a leading mobile network operator, had noticed that subscribers exchanged airtime to transfer money. M-PESA formalised this practice, and the concept expanded fast, in part thanks to the high penetration of mobile phones. By 2013, 74 per cent of the population over 15 had an account with one of four MFS providers: M-PESA, Airtel Money, yuCash, and Orange Money [18].

Today, M-PESA offers multiple services; see Table 1. The transfer function can now also be used to pay in shops, and there are mobile banking features. For instance, in 2012 Safaricom teamed up with Commercial Bank of Africa (CBA) to launch M-SHWARI, a financial services suite that includes an interest-paying bank account at CBA [19]. After disagreements between Safaricom and partner Equity Bank, an earlier similar product, M-KESHO, is no longer promoted [19] (p. 17).

Given this wide range of MFS, how did we define MFS-enabled financial inclusion? Our starting point was the broad definition of Klapper and Singer [1] quoted in the Introduction ("access to and usage of appropriate, affordable, and accessible financial services"). However, we opted for a narrow definition of 'financial services'. Unlike InterMedia [20] (p. 9), we do not consider the use of M-PESA for transfers to be sufficient for 'real' inclusion. The ability to receive remittances by way of M-PESA is an obvious improvement over past practices (see Section 3), but it does leave recipients in a dependent position. In line with the old saying that we paraphrase in our title—'Of saving comes having'—we consider the ability to save as a necessary condition for full financial inclusion. This is in line with the Committee on Payments and Market Infrastructures and the World Bank Group [10] (p. 6), and especially Johnson [21] (p. 3): "Beyond use for payments, whether or not people actively save in these [e-money] accounts is a key issue for financial inclusion". From this perspective, saving on M-SHWARI clearly qualifies. But we also consider people who save on their mobile account as financially included, even though such funds do not earn any interest. Demombynes and Thegeya [22] call this "basic mobile savings", as opposed to "bank-integrated mobile savings".

**Table 1.** Penetration of M-PESA uses (N = 2994).

| M-PESA Uses | Number | Percentage |
|---|---|---|
| Own a SIM card | 2454 | 82.0 |
| Own or have access to a SIM card | 2832 | 94.6 |
| Use M-PESA | 2171 | 72.5 |
| Use M-KESHO | 34 | 1.1 |
| Use M-SHWARI | 283 | 9.4 |
| Mobile money transfers | | |
|   Withdraw money [a] | 2303 | 76.9 |
|   Deposit money | 1868 | 62.4 |
|   Pay for goods at a store | 54 | 1.8 |
|   Receive money for regular support | 1235 | 41.2 |
|   Send money for regular support | 1118 | 37.3 |
|   Receive money for emergency | 761 | 25.4 |
|   Send money for emergency | 764 | 25.5 |
| Mobile banking | | |
|   Save money for future purchase/payment | 205 | 6.8 |
|   Receive a salary | 59 | 2.0 |
|   Take a loan | 37 | 1.2 |
|   Receive state aid or pension | 18 | 0.6 |
|   Buy insurance | 5 | 0.2 |

*Notes.* Source: [15]. [a] The number of people withdrawing money is higher than the number of M-PESA users because even non-users can receive money. See [23] (Box 1, pp. 140–144) for a more detailed description.

Table 1 shows the proportions of people who, according to the FII data that we exploit, use M-PESA for specific operations. As can be seen, basic functions are widely used. However, Kenyans seem less keen on the more evolved mobile banking services. Only 6.8 per cent of the respondents had, at the time of the survey, ever saved on their M-PESA account.

## 3. State of the Literature on M-PESA

This section provides a brief overview of the extant literature. Following [24], we have grouped the papers depending on whether they examine adoption of M-PESA, its usage, or its impact on society—in that order. Note that in this section we focus exclusively on Kenya (because both the setting and the characteristics of the MFS schemes may differ across countries [23] (p. 167)). In later sections we do, however, compare our set-up and results with approaches and findings of papers for other African countries. For geographically broader overviews, see [25] and [23].

### 3.1. Adoption

What motivates or inhibits a potential user to adopt M-PESA? Most studies focus on the role of socio-demographic variables. Porteous [26] finds that people who are less educated and are less at ease with mobile phone technology were less likely to be early adopters. Using data from the 2009 FinAccess survey, Aker and Mbiti [27] find that MFS adopters in East Africa were, at the time, mainly well-off: adoption proved to correlate positively with education, bank account possession, urbanity, and wealth. In the same line, using the same dataset, Johnson and Arnold [28] find the determinants to be similar to those for bank accounts—with one exception: age did not have a significant positive impact on M-PESA adoption. But the profile of M-PESA users did change over time. Jack and Suri [29] observe that while during the first round of their survey, in 2008, only 25 per cent of M-PESA users were unbanked and 29 per cent came from rural areas, a year later these figures were 50 and 41 per cent, respectively. The share of primary educated users and women also increased.

Morawczynski [30], for her part, analyses to what extent M-PESA adoption is linked not so much to socio-demographic characteristics but to demand for financial services, accessibility, and perceived usefulness. Using evidence for areas where traditional banking facilities are scarce, she finds that the dense network of M-PESA agents and the speed of the transfer have a strong impact. The security offered by M-PESA as a store of value also matters. In a later paper, Morawczynski and Pickens [31] (p. 2) find that urban users adopt M-PESA because it is "cheaper, easier to access, and safer than other money transfer options". They also observe a prescription effect between urban and rural users, in that the former tend to encourage their social circle to adopt the system.

### 3.2. Use

The positive economic effects of M-PESA identified below, in Section 3.3, revolve around two uses: remitting (especially in case of an emergency) and saving. To start with remittances, Jack and Suri [29] identify a pattern where urban family members support their rural relatives, a situation that probably results from a migration process. More in particular, Morawczynski and Pickens [31] observe money flows between male urban senders and female rural recipients. Johnson's 2010/2011 data for three rural areas also reveal a strong pattern of receipt of funds from spouses or children who are "sending money home", but also suggest "strong patterns of transactions with other relatives and an important though smaller role for friends" [4] (p. 91).

Turning to saving, between the two rounds of their survey Jack and Suri [29] effectively see an increase in the use of M-PESA as a savings tool, from about 75 per cent of the adopters in round 1 to 81 per cent by round 2, in 2009. Note that in their broad definition 'saving' refers to any form of keeping money in a storage means for more than 24 h. However, the incidence of longer-term saving would also appear to have risen: the percentage of households who save on M-PESA for emergencies increased from 12 to 22 per cent.

Relying on the 2009 FinAccess survey, Mbiti and Weil [32] find that, overall, 26 per cent of users save on M-PESA. As in Jack and Suri [29], the incidence is higher among the banked than among the unbanked (35 vs. 19 per cent). Demombynes and Thegeya [22] use data from a dedicated survey conducted in 2010. They define respondents as having "mobile savings" if they answer affirmatively to the question "Do you save any portion of your income?" *and* list M-PESA as one of the places where they save. 15 per cent of the respondents (33 per cent of the users) had such savings.

Kikulwe, Fischer and Qaim [33] conducted two smaller-scale surveys among 320 banana-growing farm households in the Central and Eastern provinces of Kenya. In 2010, over 40 per cent stated that they use their mobile money account as a savings tool, and about 27 per cent used it as a means of transferring money to their formal bank account. The article does not provide details as to the definition of saving. Finally, Johnson [4,21] reports on another smaller-scale survey conducted in 2010/2011 in three rural districts. Johnson finds that after receipt of a transfer the majority of the M-PESA users withdrew the funds completely. Some 34 per cent reported holding a balance on their phone.

Clearly, the large discrepancies in the propensity to save on M-PESA reported by the different studies might in part be caused by differences in the scope and representativeness of the surveys [33] (p. 12), but they also point toward a definition problem [21] (pp. 3–4). For one, as Demombynes and Thegeya [22] (p. 10) stress, responses "reflect each respondent's subjective understanding of what it means to 'save your money'". This might apply, for example, to the practice—documented by Plyler, Haas, and Nagarajan [34]—of weekly savings being spent at the end of the week. Also, in Johnson's [4] survey, the most common reason for holding a balance on one's phone was safety (19 per cent). As one respondent put it, "you can walk with the money and you don't have it" [4] (p. 92). Johnson stresses that this is "not coterminous with a place to 'save' in the sense of building up balances". Hence, "keeping money in an [M-PESA] account needs careful interpretation, especially when surveys ask about 'saving'" [21] (p. 13). As will become clear in Section 5.1, the definition of saving in the survey that we exploit in this paper is more stringent than in the papers just discussed and, as a result, its incidence is substantially lower.

### 3.3. Economic Impact

We now discuss the literature on the positive consequences of the two uses of M-PESA just documented. Where remittances are concerned, Jack and Suri [35] note that M-PESA transfers are immediate and less costly—for amounts large *and* small. This enables small emergency remittances in response to small-magnitude shocks. Moreover, the availability of M-PESA should allow households to send members further, thus optimising migrants' revenues and increasing remittance flows.

Morawczynski [36], in her qualitative study, conducts two interrelated surveys (in an urban slum and in a rural village where some of the urban dwellers migrated from), and analyses the evolution in financial behaviour as a result of the adoption of M-PESA. Morawczynski observes an increase in total remittance flows, resulting in a decrease in the financial vulnerability of the receivers. Jack and Suri [35] demonstrate quantitatively that M-PESA remittances effectively improve resilience to income shocks. They find a reduction of per capita consumption of 7 per cent among non-user households facing shocks, while consumption of households with access to M-PESA is not affected; see also Aker and Mbiti [27]. The qualitative study by Plyler et al. [34] confirms these findings: participants credited M-PESA with boosting local consumption thanks to 'rescue money'. This shock smoothing was also perceived as having increased agricultural productivity. In the same line, the regression results of Kikulwe et al. [33] suggest that mobile money use is welfare-enhancing for smallholder households, who constitute the majority of the rural poor. It not only results in higher remittances but also stimulates more commercially-oriented farming.

Beyond higher remittances, Jack and Suri [35] also predict an increase in household savings—because of the security provided by M-PESA—as well as higher investment in human and physical capital. However, as Arestoff and Venet [37] (p. 3) point out, where saving is concerned

the results are less clear-cut than for money transfers. In their probit analysis, Demombynes and Thegeya [22] find, after controlling for socio-demographic factors, that M-PESA users are 20 per cent more likely to have "savings of any kind". However, in spite of the use of instrumental variables their regressions might suffer from an endogeneity problem, in that the causality could also run from savings to M-PESA adoption. Those who report *not* having any savings *in any of the forms* are undoubtedly poorer than average. Some of them may not have access to a SIM card and are thus simply not in a position to register for M-PESA. A similar criticism can be leveled at [38].

In a recent paper, Suri and Jack [17] provide the most comprehensive evaluation of the welfare effects of M-PESA to date—in two respects. For one, they do not limit their analysis to remittances and saving, but also look into the effects of migration and changes in occupational choice. Second, they try to gauge the long-run impacts. To that end, Suri and Jack complemented the 2008 and 2009 rounds of their household panel survey—exploited in Jack and Suri [35]—with three additional rounds. To identify the causal effects of M-PESA they use changes in access to mobile money—quantified by the number of M-PESA agents within one kilometer of the household—rather than adoption itself. Specifically, they compare outcomes, as measured in the 2014 survey, of households that saw large increases in agent density between 2008 and 2010 with outcomes of households that experienced smaller increases.

Suri and Jack estimate that access to M-PESA lifted at least 194,000 Kenyan households, or 2 per cent of the total, out of poverty. They point out that the higher observed consumption levels could be driven by multiple mechanisms, and find some support for the savings and occupational choice channels. In particular, they find that an increase in M-PESA agent density positively affects total financial savings of households—although *not* savings on M-PESA itself. Also, individuals who experienced larger increases in M-PESA access were less likely to be working in farming and more likely to be working in "business or sales". Suri and Jack also provide tentative evidence concerning the financial 'stepping stone' effect of M-PESA anticipated by other authors [1,31]. Suri and Jack find that the change in access to mobile money effectively predicts the adoption of a bank account. They are, however, quick to point out that this may be driven by a supply-side response.

Finally, perhaps the most interesting result of Suri and Jack relates to the empowerment of women, which is a key issue in Kenya. Morawczynski and Pickens [31] already saw great improvement in women's independence thanks to more regular and immediate remittances. The explanation would seem to be that, in patriarchal households, the privacy offered by M-PESA as a store of value allows women to manage household liquidity with less control by the male head. Suri and Jack [17] underpin this beneficial effect: their regressions show that consumption growth, the reduction in poverty, and the switch away from subsistence agriculture are all more pronounced for female-headed households.

## 4. Data

The nationally representative household survey (N = 3000) that we exploit was conducted as part of InterMedia's first eight-country Financial Inclusion Insights Program, funded by the Bill and Melinda Gates Foundation. As explained, we focus on Kenya, for which the data pertain to 2013. The survey consisted of face-to-face interviews lasting 45 to 60 minutes. Households were selected randomly, based on the latest available census; see [39] for full details.

Table A1 in the Appendix A contains selected descriptive statistics; Table S1 in the Supplementary Materials contains the full set. Table A1 also contains statistics for the 2013 national Financial Access survey, the dataset that we use to perform robustness checks [16]. There are a number of notable differences between the two sets. Most importantly, at 72.7 per cent the possession of M-PESA accounts is substantially higher in our data than in the FinAccess data (58.6 per cent). Given the rapid growth of M-PESA, part of the explanation may lie in fact that the data collection period for the FinAccess survey was 1 October 2012–1 February 2013, whereas the FII data was collected up to one year later (between 12 September and 4 October 2013). Note also that the proportion of bank account owners in the FII

data (28.2 per cent) is lower than the 2014 figure of 55.2 per cent in the Findex database [3]. This is due to our focus on bank accounts, whereas Findex also consider accounts at other financial institutions.

Turning to gender, it is clear that, at 62 per cent, women are overrepresented: the national gender ratio for Kenya in 2011 was 1 [40]. The FinAccess data has a similar bias towards women. The age distribution is also different from official numbers (for 2014) [40], with an under-representation of the young. These differences might result from the fact that 64 per cent of the interviews were conducted between Monday and Friday, probably in part at times when many men and youngsters were at work or at school. FSD Kenya & Central Bank of Kenya [16] (p. 6) offers a similar explanation for the FinAccess data. These biases call for caution whenever penetration levels are mentioned, as in our Figures 1–4, but should not affect the results of our logit analysis as such.

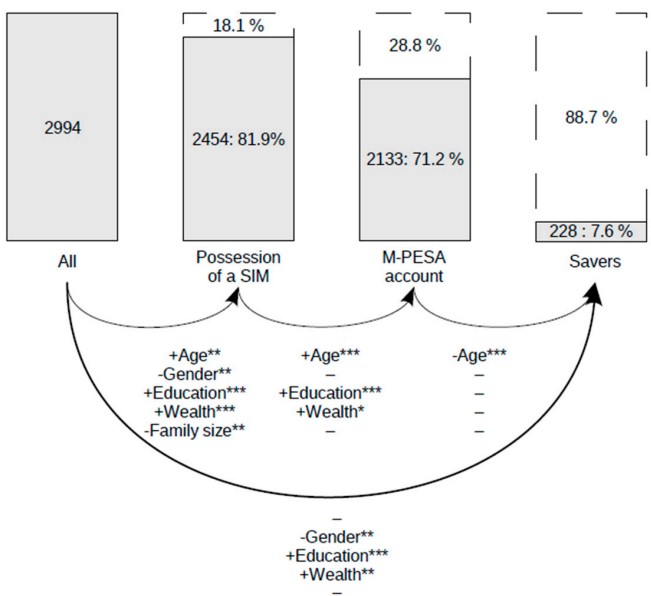

**Figure 1.** Overview of results: Urban + Rural.

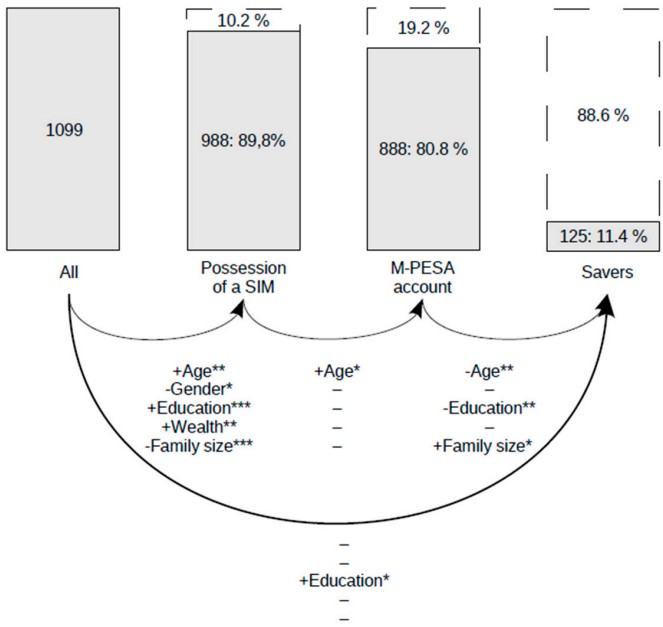

**Figure 2.** Overview of results: urban sample.

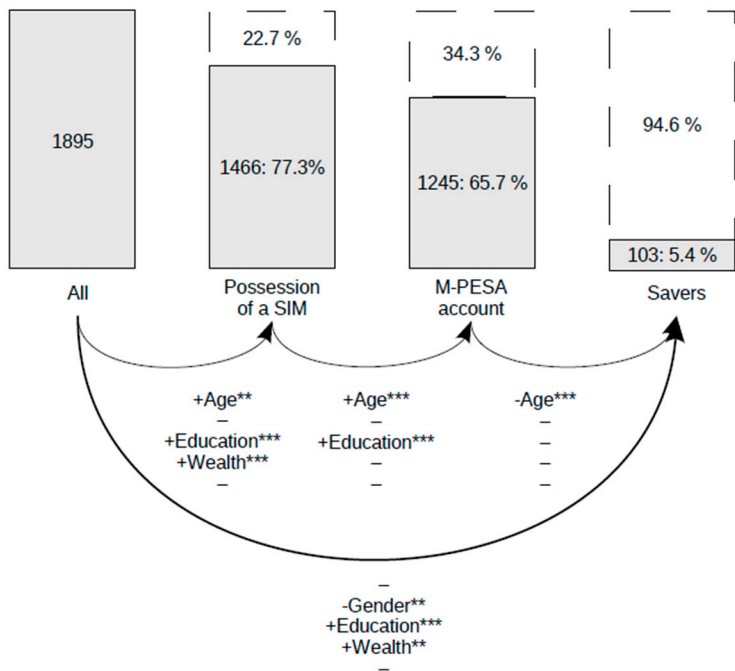

**Figure 3.** Overview of results: rural sample.

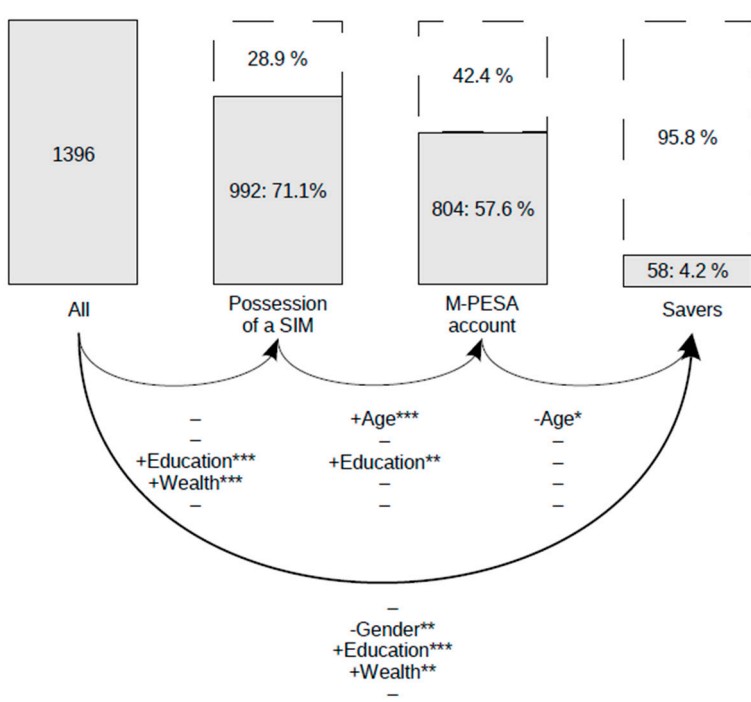

**Figure 4.** Overview of results: rural, vulnerable.

## 5. Methodology

While our main goal is to explain whether or not a respondent has used M-PESA as a savings tool, we do not just perform a straightforward probit analysis on a binary variable. Rather, we also examine two crucial preconditions. The result is a three-step approach in which we examine, first, which Kenyans own a SIM card; second, which of these individuals have an M-PESA account; and, third, which M-PESA users actually save on this account.

To the best of our knowledge, this is a novel approach. In the literature, the dominant method consists in using dummies—for both steps two and three of our approach (see also [23]). Johnson and Arnold [28] (pp. 741–742) have in their regressions for M-PESA adoption—i.e., step two of our approach—dummies for owning and having access to a phone. Kikulwe et al. [33] (pp. 7) and Munyegera and Matsumoto [41] (p. 130)—in a paper on Uganda—also use a dummy for mobile phone ownership. Murendo, Wollni, de Brauw, and Mugabi [42], in another paper on Uganda, use the number of mobile phones owned by the household. We will show that the use of dummies for mobile phone ownership hides a number of interesting relationships between variables.

Where step three of our approach is concerned, Munyegera and Matsumoto [43] (p. 51) have in their probit regression for the saving behaviour of Ugandan households a dummy variable that takes a value of 1 if the household has at least one member who use mobile money services. They do not initially control for ownership of a SIM of a mobile phone—so that the non-use of mobile money services can have very diverse causes—but afterwards do conduct propensity score matching so as to examine the effect of mobile money adoption for comparable user and non-user households.

In a paper on Ghana, Apiors and Suzuki [44] (p. 6) use a slightly different approach compared to the above: they include only mobile phone owners in their sample (and, when analysing outcome variables such as saving and consumption, include a dummy for mobile money use—as [43] do). Clearly, this approach allows to examine only steps two and three of our set-up. Crucially, one should refrain from generalised conclusions such as "financial status is not a significant indicator in determining who participates in mobile money" [44] (p. 6)—because 'financial status' may well have an impact on mobile phone ownership in (the invisible) step one [45] (see also [23] (p. 172)). Finally, there are also authors who simply ignore the preconditions: in their regressions for the saving behaviour of Kenyan households, Ouma et al. [38] do not control for ownership of a mobile phone or a SIM card.

In addition to our three-step approach, our scope also varies in another respect. At each of the above steps, we first analyse the full sample but then zoom in on the rural population and, finally, on the more vulnerable households among the latter. Note in this respect that Wyche and Olson [46] (p. 36) stress that "[ICT for development] as a field has primarily been concerned with urban populations" and that "Kenya, like most Sub-Saharan African countries, has a substantial urban-rural divide that affects ICT access". Initially we defined our third subsample as inhabitants of rural areas who do not own a bank account and who never send remittances; the idea being that both indicators are signs of (relative) financial health. However, upon closer inspection of the ethnographic literature on the topic, this assumption proved wrong for the second indicator, in that even the poor at times make use of M-PESA to provide financial help to friends and relatives. In their analysis of twelve family networks, Kusimba, Yang, and Chawla [47] (p. 1) find that money transfers are "small and frequent" and that money networks are often reciprocal: "People were not only senders or receivers, but rather, participants in groups who circulate value—groups of siblings who pool resources for a father's medical needs . . . or community members who contribute to funerals" [47] (p. 1). Therefore, we decided not to exclude respondents who send *and* receive, but only those who send but do not receive. Of the 1895 in the rural sample, 444 were excluded because they have a bank account and an additional 55 because they send remittances but do not receive any. Interestingly, Kusimba et al. [47] (p. 7) find that nodes who are only senders "are always urban workers or international migrants".

An important methodological remark is that because working with non-random subsamples introduces a risk of selection bias, we use a two-stage Heckman procedure for steps 2 and 3 of our three-step approach. The intuition is that, say, the subsample of owners of a SIM may have specific characteristics that influence possession of an M-PESA account. Therefore, in a first stage (the selection equation) we explain the probability that a respondent owns a SIM and only in a second stage (the outcome equation), the probability that he or she has an account. Similar remarks hold for step 3. A final methodological note is that, to the extent possible, we replicate all our regressions with FinAccess data for the same year. Let us now explain our variables.

*5.1. Dependent Variables*

As mentioned, we use three dependent variables: ownership of a SIM, adoption of M-PESA, and, finally, use of M-PESA as a savings tool. All three are on the individual level and are of a binary nature. The first variable takes a value of 1 when the respondent owns one or more SIM cards. Note that for having an M-PESA account—and receiving remittances via this channel—it is apparently sufficient to be able to use somebody else's SIM: of the 378 respondents who do not own a SIM but have access to one (see Table 1), 38 reported having an M-PESA account. However, it is clearly an entirely different matter to entrust somebody else (or their SIM) with your savings; there are barely any respondents who save on MFS without *owning* a SIM: 3 in the full sample and 2 in the more narrow samples. While we could thus have used access to a SIM (rather than ownership) as the selection criterion in the second step of our analysis, this would have made little sense in view of our ultimate goal. A drawback of using SIM ownership as the selection criterion is that our analysis of M-PESA adoption in step 2 is not entirely accurate as a stand-alone analysis: it omits the respondents who do not personally own a SIM but do have an account. However, the bias is limited: for the full sample, we are talking about 1.3 per cent of the total number of adopters.

Our second dependent variable, adoption of M-PESA, is self-explanatory. For the saving variable in step 3 we exploit questions where M-PESA users were asked whether or not they had already saved money for a future purchase or payment on, respectively, M-PESA, M-SHWARI, M-KESHO, or on a bank account. For saving on M-PESA, the question was: "Have you ever used a mobile money account to do the following? [...] Save money for a future purchase or payment". To be clear: the survey did not inquire about saving for other purposes.

As we are interested in the impact of M-PESA on financial inclusion, we decided to merge, in a variable 'saving on MFS', the categories 'saving on M-PESA' (that is, on the 'mobile money account') and 'saving on M-SHWARI', even though the first type of saving is non-interest bearing whereas the second is. To be clear: (1) nobody saved on an M-KESHO account, and (2) the variable 'saving on MFS' (228 respondents, 7.6 per cent of the sample, 10.5 per cent of M-PESA users) is in practice dominated by respondents who save on M-PESA but not on M-SHWARI; these respondents number 189, equivalent to 82.5 per cent of all MFS savers.

As already announced at the end of Section 3.2, because of the more stringent definition (saving *for a future purchase or payment*) the incidence of saving in our dataset—10.5 per cent of the M-PESA users—is markedly lower compared to the headline figures reported in earlier studies. But on closer scrutiny it can be reconciled with the more detailed results reported by Johnson et al. [48] and Johnson [4,21]. In Johnson's 2010/2011 survey, the reasons given for holding a balance on one's phone that would qualify as 'saving' under our definition all run in the low single digits [48] (p. 25).

*5.2. Independent Variables*

Demographic variables allow to understand the social conditions of the respondents. We choose the ones that are commonly used in the studies that opt for the socio-individual approach (see Section 3.1), namely: age, gender, wealth, education level, geographical situation (urban or rural), and family size. Table A1 provides descriptive statistics.

Our choice of age phases was driven by the possible association with motivations to save. The life cycle hypothesis typically assumes that the propensity to save follows a hump-shaped curve [38] (p. 31), with people saving more in the run-up to retirement. However, it remains to be seen whether this is a realistic assumption in the Kenyan context. Only a minority of the adult population hold a regular job and there is evidence that in order to prepare retirement Kenyans prefer to invest rather than save on an account that yields a negative real return [49].

Turning to wealth, in the FII survey respondents are not asked about their income or financial assets. However, the survey does comprise the Poverty Probability Index (PPI, formerly known as the Progress out of Poverty Index). The PPI is a country-specific indicator, introduced by the Grameen Foundation, based on ten closed questions about easily observable household characteristics.

According to a recent validity assessment for Rwanda [50], the PPI can accurately distinguish poor from non-poor households. However, we refrained from using PPI because of high correlations with our family size (−0.65) and education (0.54) variables, and to a lesser extent also with urbanity. We therefore focussed on questions 7–10 of the scorecard for Kenya [51], which inquire about the possession of essential tools: namely, an iron (0 or 1) and the number of, respectively, mosquito nets, towels, and frying pans (each time: "0", "1", "2 or more"). We used this information to build an index that ranges between 0 (poorest) and 25 (wealthiest), based on the scores that the PPI attributes to the answers. (For example, possession of one towel corresponds to 6 points.) Our 'wealth' variable is thus of a completely different nature compared to studies for the developed world. It is more of an inverted poverty index. Finally, for education we look at the highest level of schooling *completed* by the respondent. We have also experimented with a literacy variable, but this was, at best, only marginally significant.

## 6. Results

As announced, we first examine our respondents' possession of a SIM card. In Sections 6.2 and 6.3 we then focus on, respectively, the adoption of M-PESA in general (that is, for any type of use), and the use of M-PESA as a savings tool. Section 6.4 summarises the results across the three steps.

### 6.1. Precondition: SIM Ownership

Table 2 presents our regressions for ownership of a SIM card. As explained in Section 5.1, in steps 2 and 3 of our analysis we will use SIM *ownership* as the selection criterion rather than access, as only few respondents who do not own a SIM have an M-PESA account, let alone save on it. If we focus first on the full sample, we can see in regression (1) that, overall, those who are older than 25, male, educated, and are part of a 'wealthy' (in fact: non-poor) household have a higher probability of owning a SIM card. When added, urbanity also has a very significant positive effect. Conversely, the higher the family size, the lower the probability of owning a SIM.

An examination of the average marginal effects (AMEs) reveals, first of all, that the higher the education level, the stronger the effect: completion of primary education increases the probability of owning a SIM card by 13 percentage points compared to the non-educated, completion of secondary education increases it by 23 percentage points, and literally all respondents who have completed university own a SIM card; hence the 'no variation' in the Table. Turning to age, here, too, the effect increases in size as we move up the categories, but only up to a point (namely the 36–40 age bracket). Note in particular that while those over 55 do have a higher probability to own a SIM card than the 15–25 age group, at 4.5 percentage points the AME is substantially smaller than for all other age ranges (for which the AMEs lie between 10 and 13 percentage points). This might be an indication of technology aversion.

In regressions (3) and (5), we progressively zoom in on the more vulnerable groups in the Kenyan population. The rural population is vulnerable to weather hazards—which translate into volatile food prices and uncertain incomes—and is thus, on average, probably less able to cope with shocks. However, we noticed that some respondents in the rural sample do own a bank account, which is a sign of (relative) wealth. Therefore, we excluded these individuals from the 'rural, vulnerable' subsample, along with—as explained in Section 5.1—those who send remittances but never receive any.

Overall, the results of regressions (3) and (5) are similar to those for the full sample. Importantly, in line with the results of Murphy and Priebe [52] (p. 5), those who do not own a SIM are predominantly non-educated and poor. Note also that the AMEs of education and wealth are bigger for the 'rural, vulnerable' sample (respectively 14.0 and 1.8 percentage points) than for the urban sample (7.2 and 0.4). A less intuitive difference is that both gender and family size lose their significance. In particular, one would expect access to a phone to be more of problem for rural than for urban women. We come back to this below.

**Table 2.** Possession of a SIM card.

| | Urban + Rural (1) | Urban (2) | Rural (3) | Rural (4) | Rural, Vulnerable (5) |
|---|---|---|---|---|---|
| Age | ** | ** | ** | n.s. | n.s. |
| 15–25 | – | – | – | – | – |
| 26–30 | 0.454 *** | 0.595 *** | 0.392 ** | 0.394 ** | 0.381 ** |
| | (4.63) | (3.55) | (3.15) | (3.24) | (2.91) |
| 31–35 | 0.480 *** | 0.489 ** | 0.497 *** | 0.419 *** | 0.485 *** |
| | (4.57) | (2.69) | (3.80) | (3.33) | (3.54) |
| 36–40 | 0.597 *** | 0.638 ** | 0.599 *** | 0.503 *** | 0.526 *** |
| | (5.43) | (2.95) | (4.57) | (4.00) | (3.74) |
| 41–55 | 0.521 *** | 0.640 *** | 0.534 *** | 0.427 *** | 0.457 *** |
| | (5.77) | (3.51) | (4.91) | (4.09) | (3.91) |
| Over 55 | 0.203 * | 0.350 | 0.226 * | −0.00383 | 0.166 |
| | (2.13) | (1.68) | (2.01) | (−0.04) | (1.33) |
| Gender (Female = 1) | −0.165 ** | −0.326 * | −0.116 | −0.201 ** | −0.103 |
| | (−2.62) | (−2.44) | (−1.59) | (−2.84) | (−1.28) |
| Education | *** | *** | *** | | *** |
| Non-educated | – | – | – | | – |
| Primary | 0.587 *** | 0.453 ** | 0.568 *** | | 0.546 *** |
| | (8.59) | (3.22) | (7.02) | | (6.15) |
| Secondary | 1.049 *** | 0.916 *** | 1.011 *** | | 0.865 *** |
| | (10.77) | (5.36) | (7.94) | | (5.87) |
| College | no variation | no variation | no variation | | no variation |
| Wealth | 0.0542 *** | 0.0292 ** | 0.0633 *** | 0.0833 *** | 0.0609 *** |
| | (9.83) | (2.73) | (9.62) | (13.54) | (8.43) |
| Family size | −0.0316 ** | −0.0881 *** | −0.00953 | −0.0244 * | −0.00922 |
| | (−3.11) | (−3.57) | (−0.82) | (−2.18) | (−0.74) |
| Constant | −0.295 * | 0.593 *** | −0.606 *** | −0.299 * | −0.637 *** |
| | (−2.57) | (2.59) | (−4.38) | (−2.29) | (−4.30) |
| Pseudo $R^2$ | 0.1685 | 0.1273 | 0.1708 | 0.1268 | 0.1376 |
| AIC | 2351.6 | 642.4 | 1694.0 | 1788.1 | 1469.9 |
| BIC | 2417.4 | 697.0 | 1754.9 | 1838.0 | 1527.6 |
| Log likelihood | −1164.8 | −310.2 | −836.0 | −855.0 | −723.9 |
| Observations | 2994 | 1099 | 1895 | 1895 | 1396 |

*t* statistics in parentheses; *, ** and *** indicate significance at the 0.05, 0.01 and 0.001 level, respectively; asterisks next to the name of a categorical variable indicate its overall significance.

As announced, in a robustness check we have tried to replicate all our regressions with 2013 FinAccess data. A problem for the regressions in Table 2 was that the FinAccess survey does not inquire about ownership of a SIM, but only about ownership of a phone. Note that this also has implications for steps 2 and 3 below, in that the Heckman selection criteria will be stricter. This said, the results in Table S2 in the Supplementary Materials are, apart from the precise values of the coefficients, almost a carbon copy of Table 2. For one, we again find very significant positive effects for age, education, and wealth—and almost identical patterns across age and education categories. Moreover, there is again a negative gender and family size effect in the full sample but not among the 'rural, vulnerable'; a difference is the significant gender effect for the rural sample. Overall, these results chime with Blumenstock and Eagle's [53] results for Rwanda, in 2009. They also find that mobile phone owners were wealthier, better educated, and predominantly male. A difference is that they find that phone owners come from larger households.

From a methodological angle, let us stress that, in their paper on Uganda, Murendo et al. [43] also find that poor households are less likely to own a phone, but only indirectly. They find that the number of mobile phones owned has a significant positive impact on the adoption of mobile money, and when they split their sample into poor and non-poor households it becomes clear that the impact is bigger for poor households [43] (p. 338, Table 5). Our three-step approach has the advantage of directly uncovering such relationships.

Upon closer inspection, the negative impact of gender and family size in the full sample in both Table 2 and Table S2 is caused by the urban population; see regression (2) in Table 2. Part of the explanation for why an increase in family size reduces the probability that a household member owns a SIM/phone among the urban but not among the rural population might lie in the significantly higher incidence of singles in cities (17.3 vs. 7.3 per cent; *t* = 2.8; *p* = 0.00). Many of these are probably migrants who need to stay in contact with their family on the countryside. When we replaced family size by a dummy that takes a value of 1 for single-person households, this dummy was positive and significant at the 1 per cent level in both the full and the urban sample, but not in the rural samples (results not reported). Where gender is concerned, an important preliminary remark is that SIM ownership among rural women (71.4 per cent) is effectively lower than among their urban counterparts (82.6 per cent). However, this is not so much because the gender effect as such is stronger, but rather because rural households are poorer than urban households (mean wealth is 13.5 vs. 15.1; *t* = 8.6; *p* = 0.00), so that rural men also have a lower probability to own a SIM. Note that no less than 91.2 per cent of the urban respondents have a SIM, compared to only 77.4 per cent of the rural. Within the rural sample, the lower education of women (0.73 vs. 0.87 for men; *t* = 9.5; *p* = 0.00) also plays a role. As can be seen in regression (4), when education is removed, gender becomes significant.

## 6.2. M-PESA Adoption

In Table 3 we now explain the possession of an M-PESA account by those respondents who are in a position to open one; that is, who own a SIM card. One could argue that we should restrict our sample even more, namely to respondents who own a SIM of Safaricom, the mobile operator behind M-PESA. However, of the respondents who own a SIM, 94 per cent have a Safaricom SIM (either exclusively or as one of their SIMs). And respondents who own a SIM from another operator could probably also afford a Safaricom SIM. As a point of comparison, we also explain the possession of a bank account. Regressions (4) and (5) again zoom in on the more vulnerable population groups.

As explained in Section 5, from this step onwards we use a two-stage Heckman procedure. A key requirement for the use of such a model is that the selection equation contains at least one (significant) variable that does not appear in the outcome equation. Therefore, we have included 'Employed', a dummy variable that takes a value of 1 when the respondent holds a regular job, in the selection equation. A drawback is that multicollinearity—with, for example, 'Age'—can affect the coefficients of the other socio-demographic variables in the selection equation. But then the purpose of the selection equation is not to identify the determinants of SIM ownership—this being done in the first step—but simply to obtain a good prediction. Reassuringly, all goodness-of-fit indicators of regressions for SIM ownership with and without 'Employed' were just about identical.

If we then discuss the results in the order in which the variables appear in Table 3, for age the general picture is one where the higher age ranges have a higher probability of owning an M-PESA account. The precise pattern does differ across the samples. For the urban subsample, age is overall even only significant at the 5 per cent level. Note that Johnson and Arnold [28] (p. 741), who enter age as a continuous variable, do not find a significant effect of age. Our result might surprise, in that the young are typically the early adopters of new technologies. However, the innovation we examine in this paper has a crucial financial dimension. Many of the 15- to 25-year-olds, for example, may still depend on their parents. Even in developed countries, therefore, it is not exceptional to see that the youngest do not use the latest in payment technologies. For example, where The Netherlands is concerned, Jonker [54] finds that younger consumers between the age of 15 and 24 are the heaviest cash users, even more so than those over 65.

Turning to gender, in line with Johnson and Arnold [28] (p. 742) we find no significant effect on M-PESA adoption. There is, however, a gender effect for the possession of a bank account, in regression (2). This is plausible in a patriarchal society such as Kenya, where in most households the male head will be the owner of the account. Dupas and Robinson [55] reveal that in rural areas 10 per cent of the women have a bank account while for men this number is 21 per cent. For education,

we find, as expected and again in line with [28], that, apart from the urban sample, the educated have a higher probability of owning an M-PESA account. Kikulwe et al. [33] (p. 7, Table 3) find a significant positive effect of the education level of the household head. Note also that the AMEs (not reported) increase with the education level. Both observations also hold for the possession of a bank account in model (2). To be clear: in model (5), all respondents who have completed college own an M-PESA account.

**Table 3.** Possession of formal accounts by individuals who own a SIM card.

| | Urban + Rural | | Urban | Rural | Rural, Vulnerable |
|---|---|---|---|---|---|
| | M-PESA (1) | Bank Account (2) | M-PESA (3) | M-PESA (4) | M-PESA (5) |
| **Outcome Equation** | | | | | |
| Age | *** | *** | * | *** | *** |
| 15–25 | – | – | – | – | – |
| 26–30 | 0.202 * | 0.424 *** | 0.334 * | 0.122 | 0.105 |
| | (2.10) | (4.50) | (2.17) | (0.95) | (0.74) |
| 31–35 | 0.319 ** | 0.440 *** | 0.454 * | 0.256 | 0.283 |
| | (2.92) | (4.36) | (2.37) | (1.80) | (1.68) |
| 36–40 | 0.415 *** | 0.618 *** | 0.796 ** | 0.320 * | 0.354 * |
| | (3.58) | (5.90) | (2.95) | (2.27) | (2.08) |
| 41–55 | 0.428 *** | 0.748 *** | 0.287 | 0.483 *** | 0.512 ** |
| | (4.21) | (7.58) | (1.54) | (3.59) | (3.09) |
| Over 55 | 0.441 *** | 0.737 *** | 0.222 | 0.510 *** | 0.607 ** |
| | (3.73) | (6.43) | (0.92) | (3.41) | (3.09) |
| Gender (Female = 1) | −0.0527 | −0.422 *** | −0.109 | −0.0496 | 0.0123 |
| | (−0.78) | (−6.77) | (−0.84) | (−0.60) | (0.13) |
| Education | *** | *** | n.s. | *** | ** |
| Non-educated | – | – | | – | – |
| Primary | 0.156 | 0.134 | 0.0521 | 0.159 | 0.189 |
| | (1.94) | (1.35) | (0.27) | (1.64) | (1.56) |
| Secondary | 0.456 *** | 0.764 *** | 0.354 | 0.456 *** | 0.607 ** |
| | (4.50) | (5.55) | (1.46) | (3.48) | (3.18) |
| College | 0.728 * | 1.951 *** | 0.334 | 4.096 | no variation |
| | (2.25) | (7.31) | (0.82) | (0.03) | |
| Wealth | 0.0131 * | 0.0191 * | 0.0141 | 0.0135 | 0.00820 |
| | (2.05) | (2.57) | (1.13) | (1.64) | (0.78) |
| Family size | −0.0108 | −0.0160 | −0.0251 | −0.00431 | 0.000427 |
| | (−0.89) | (−1.35) | (−0.82) | (−0.33) | (0.03) |
| **Selection Equation** (SIM ownership) | | | | | |
| Age | 0.0390 * | 0.0388 * | 0.0936 ** | 0.0366 | 0.0212 |
| | (2.28) | (2.26) | (2.69) | (1.80) | (0.95) |
| Gender | −0.102 | −0.108 | −0.286* | −0.0530 | −0.0574 |
| | (−1.63) | (−1.72) | (−2.12) | (−0.73) | (−0.72) |
| Education | 0.548 *** | 0.548 *** | 0.461 *** | 0.535 *** | 0.475 *** |
| | (12.13) | (12.06) | (5.60) | (9.53) | (7.36) |
| Wealth | 0.0539 *** | 0.0528 *** | 0.0305 ** | 0.0619 *** | 0.0598 *** |
| | (9.96) | (9.70) | (2.87) | (9.52) | (8.34) |
| Family size | −0.0272 ** | −0.0245 * | −0.0810 ** | −0.00792 | −0.00841 |
| | (−2.73) | (−2.42) | (−3.27) | (−0.68) | (−0.66) |
| Employed | 0.236 *** | 0.255 *** | 0.210 | 0.232 ** | 0.237 ** |
| | (3.81) | (4.03) | (1.82) | (3.07) | (2.97) |
| athrho | −1.038 | −0.622 ** | −0.395 | −1.070 | −1.028 |
| | (−1.83) | (−2.59) | (−0.39) | (−1.77) | (−1.35) |
| Chi$^2$ | 5.66 | 4.13 | 0.15 | 4.96 | 2.96 |
| *p* | 0.0173 | 0.0422 | 0.69 | 0.02 | 0.08 |
| Log likelihood | −2069.6 | −2484.2 | −623.6467 | −1416.378 | −1176.13 |
| Censored | 540 | 540 | 111 | 429 | 404 |
| Uncensored | 2454 | 2454 | 988 | 1466 | 992 |

*t* statistics in parentheses; *, ** and *** indicate significance at the 0.05, 0.01 and 0.001 level, respectively; asterisks next to the name of a categorical variable indicate its overall significance.

Perhaps somewhat surprisingly, while 'wealth' is positively correlated to adoption it is only significant in the full sample. But then one should not forget that the selection criterion is SIM ownership. In other words, the poorest have already been left behind. Note that wealth is very significant in Table 2. Finally, family size is not significant.

To complete the profile of M-PESA users, it can be noted that, when introduced in models (1) and (2), urbanity is positive and highly significant (not reported). The same is true if possession of a bank account is added to models (1) and (4). This is not surprising given that regression (2) shows that the possession of a bank account is explained by the same variables as the possession of an M-PESA account. We find that 91 per cent of the banked population also use M-PESA, compared to 65 per cent of the unbanked. This is a strong indication that M-PESA and banks are complementary, at least for the bank account owners. Arestoff and Venet [37] (p. 4) call this the "additive model" of mobile banking: "People already have a bank account and new financial services become available through their mobile phone". We are more interested in the "converted model", where people (especially the poor) replace informal by formal services.

However, the relative importance of the unbanked among the M-PESA users has clearly increased over the years. Jack and Suri [35] found that the unbanked accounted for 25 per cent of M-PESA users in 2008, and 50 per cent in 2009. We find that in 2013, at least 64 per cent of M-PESA users were previously unbanked. In short, even though higher education, bank account possession, and urbanity still significantly increase the probability of owning an M-PESA account—three results that are in line with Aker and Mbiti [27]—we notice a broader uptake of M-PESA.

Finally, the robustness check with FinAccess data in Table S3 yields broadly similar results, but there are a number of differences in the regressions for possession of an M-PESA account: the coefficients of the lower age ranges are more significant (and the AMEs are bigger), wealth is significant for the urban sample, and—above all—gender now has a positive sign. This may come as surprise, but one should keep in mind that the selection criterion is substantially stricter compared to Table 3, namely ownership of a phone rather than a SIM. As a result, the poorest are not in the sample, and *among the phone owners* (rural) women are apparently not disadvantaged, on the contrary even (the AME is 3.5 percentage points).

*6.3. Saving*

In the third step of our analysis, we examine which factors affect M-PESA users' propensity to save on their M-PESA and/or M-SHWARI account. As mentioned in the literature review, saving can be understood in many ways. The definition used in the FII survey (saving *for a future purchase or payment*; see Section 5.1) is rather specific and of a longer-term nature. By contrast, the FinAccess survey leaves it to the respondents to determine exactly what constitutes saving. The question was: "Which of the following things are you currently using mobile money (M-PESA/Airtel Money/YuCash /Orange Money) for?" —with one of the possible answers being "To save".

In Table 4 the standout observation is that just about all variables now have a negative sign. This is also true in Table S4, where we perform the same analysis with FinAccess data. For age an overall negative correlation is not surprising. As explained in Section 5.2, a decrease in the propensity to save after retirement is only normal. However, we also find significant negative coefficients for lower age categories (albeit not the same in Table 4 and Table S4). This may be an indication that these respondents are busy investing (in good houses, land, livestock, etc.) rather than saving. In the terminology of Johnson and Krijtenburg [49], they would rather 'make their money work' than putting it in an account that yields a negative real return.

**Table 4.** Saving on MFS – M-PESA users.

| | Urban + Rural (1) | Urban (2) | Rural (3) | Rural, Vulnerable (4) |
|---|---|---|---|---|
| **Outcome equation** | | | | |
| Age | *** | ** | *** | * |
| 15–25 | – | – | – | – |
| 26–30 | −0.0484 | −0.0835 | −0.0360 | 0.216 |
| | (−0.54) | (−0.85) | (−0.30) | (0.89) |
| 31–35 | −0.186 | −0.394 ** | −0.0563 | 0.0753 |
| | (−1.82) | (−2.72) | (−0.41) | (0.28) |
| 36–40 | −0.121 | −0.0886 | −0.185 | −0.100 |
| | (−1.13) | (−0.61) | (−1.42) | (−0.38) |
| 41–55 | −0.265 ** | −0.232 | −0.332 ** | −0.147 |
| | (−2.59) | (−1.62) | (−2.66) | (−0.54) |
| Over 55 | −0.529 *** | −0.400 * | −0.576 *** | −0.580 * |
| | (−4.20) | (−2.14) | (−3.89) | (−2.03) |
| Gender (Female = 1) | −0.0783 | 0.0182 | −0.108 | −0.278 |
| | (−0.98) | (0.16) | (−1.15) | (−1.69) |
| Education | n.s. | ** | n.s. | n.s. |
| Non-educated | – | | – | – |
| Primary | −0.0327 | −0.0362 | −0.214 | −0.0829 |
| | (−0.18) | (−0.19) | (−1.25) | (−0.22) |
| Secondary | −0.0678 | −0.123 | −0.301 | 0.102 |
| | (−0.24) | (−0.43) | (−1.15) | (0.16) |
| College | −0.666 | −0.826 * | −0.697 | no variation |
| | (−1.84) | (−2.37) | (−1.61) | (−0.00) |
| Wealth | −0.0122 | −0.0118 | −0.0214 | −0.00102 |
| | (−1.05) | (−1.15) | (−1.53) | (−0.03) |
| Family size | 0.00785 | 0.0460 * | 0.00487 | 0.00681 |
| | (0.53) | (1.97) | (0.32) | (0.28) |
| **Selection equation** | | | | |
| Age | 0.0828 *** | 0.102 *** | 0.0965 *** | 0.0954 *** |
| | (5.25) | (3.47) | (5.05) | (4.41) |
| Gender | −0.104 | −0.189 | −0.0926 | −0.0492 |
| | (−1.88) | (−1.82) | (−1.40) | (−0.64) |
| Education | 0.520 *** | 0.385 *** | 0.542 *** | 0.548 *** |
| | (13.48) | (5.80) | (11.08) | (9.23) |
| Wealth | 0.0489 *** | 0.0270 ** | 0.0576 *** | 0.0557 *** |
| | (10.10) | (3.01) | (9.75) | (8.34) |
| Family size | −0.0318 ** | −0.0681 ** | −0.00572 | −0.00590 |
| | (−3.18) | (−3.29) | (−0.51) | (−0.46) |
| Employed | 0.245 *** | 0.188 | 0.238 ** | 0.261 ** |
| | (4.16) | (1.86) | (3.10) | (3.04) |
| athrho | −1.114 * | −1.608 | −1.594 ** | −0.923 |
| | (−2.30) | (−1.55) | (−2.66) | (−1.06) |
| Chi$^2$ | 4.47 | 2.44 | 5.64 | 0.81 |
| *p* | 0.0345 | 0.1179 | 0.0175 | 0.3680 |
| Log likelihood | −2218.225 | −838.3561 | −1345.318 | −1005.425 |
| Censored | 861 | 211 | 650 | 592 |
| Uncensored | 2133 | 888 | 1245 | 804 |

*t* statistics in parentheses; *, ** and *** indicate significance at the 0.05, 0.01 and 0.001 level, respectively; asterisks next to the name of a categorical variable indicate its overall significance.

The negative (but insignificant) coefficients for wealth point in the same direction, and the same is true for education. In Table 4, education is only significant in the urban sample (overall and for the 'college' category), but the results are substantially stronger for the FinAccess data in Table S4, where, to reiterate, the selection is stricter (in that respondents who do not own a phone are eliminated). In Table S4, education is negative and significant in the full, urban, and rural samples, but interestingly

not among the 'rural, vulnerable'. Note also that, overall, the AMEs increase with the education level: in the rural sample, for example, completion of primary education lowers the probability of saving on MFS by 8.3 percentage points compared to the non-educated; a secondary and a college degree lower it by 14.4 and 20.6 percentage points, respectively. The overall picture that emerges is one whereby those who are, unlike the 'rural, vulnerable', in a position to save on MFS—the better educated, the phone owners—are less likely to do so. This raises the question to what extent M-PESA responds to Kenyans' saving needs. We come back to this in Section 7. On a final note, let us point out that while gender has its usual negative sign, it is not significant in either datasets. Note that Demombynes and Thegeya [22] find that men are more likely to have savings of any kind (not just on their mobile account).

*6.4. A View Across the Three Steps*

Figures 1–4 summarise the results obtained for the FII data and presented in Sections 6.1–6.3. Figures S1–S4 do the same for the FinAccess data. The Figures should be read as follows. The first bar represents the total number of respondents in the (sub)sample studied. The bars to the right then indicate, for each of the steps, how many overcome the adoption barrier and how many are 'left behind'. For example, in Figure 3 for the rural population the second bar indicates that 77.3 per cent of the 1895 individuals in the subsample, or 1466, own a SIM card, whereas 22.7 per cent do not (and are thus already financially excluded at this early stage). In the next step, we then analyse which of the remaining 1466 have an M-PESA account, and so forth. As can be seen, eventually we are left with 103 respondents (or 5.4 per cent of the total) who save on MFS. At each stage, the small arrows beneath the bars indicate which socio-demographic characteristics explain the split-up in the next step. In other words, the arrows summarise the results of, respectively, Tables 2–4.

The rationale behind the bigger arrows is similar, yet different. These arrows summarise the results that are presented in Table A2 in the Appendix A. In this table, we throw overboard our three-step approach and rather explain directly, for the full (sub)samples (that is, irrespective of whether the respondents own a SIM or an M-PESA account), who saves on MFS or not. The goal here is primarily to highlight the value-added of our three-step approach. Indeed, as we illustrate below, because an explanatory variable can be crucial in one step but not in the next or, stronger, because the direction of its impact can differ between steps, there may be little or no trace of its influence in a direct estimation as in Table A2. For example, for both the FII and FinAccess data, age consistently has a significant positive impact in (one of) the first two steps but a negative impact in step three, so that its effect is absent in the direct analysis. The results for gender also differ between the step-wise and the direct approach.

In terms of methodology, the latter observation may help understand why Johnson et al. [56] and Johnson and Arnold [28] find different results for gender. Johnson et al. [56] (p. 21) find that "being a man significantly increases the likelihood of access [to an M-PESA account] compared to being a woman—*a result we did not find in the analysis of the FinAccess 2009 data where gender was insignificant*" (emphasis added). While the two papers use different samples collected in different years, the explanation might lie in the fact that, unlike Johnson et al. [56], Johnson and Arnold [28] (pp. 741–742) have in their regressions dummies for, respectively, owning and having access to a phone. These dummies, which have the biggest marginal effects on adoption of all the variables, probably capture the lower probability of women to own or have access to a phone and cause gender itself to be insignificant. Explained differently, Johnson et al. [56], who do not control for phone ownership or access, in fact lump together steps 1 and 2 of our analysis, and the gender effect that they find probably relates mostly to exclusion in step 1 rather than step 2. Conversely, Johnson and Arnold [28] do control for phone ownership/access and therefore have a cleaner analysis of step 2. In our view, this is another illustration of the value added of our three-step approach.

Turning to the actual analysis, a first interesting observation is that, even though the selection criterion is different, the results for step one are quite similar across the two datasets. There are obviously differences in significance levels and AMEs, but the only instance where a variable is

significant in one set and not in the other relates to gender in the rural sample: gender is not significant with the FII data (Table 2) but has a significant negative impact with the FinAccess data (Table S2). As already pointed out at the end of Section 6.2, this can be explained by the difference in selection criterion: the comparison across the two datasets highlights that rural women are not less likely to own a SIM, but are less likely to own a phone. With this observation in mind it is particularly interesting to have a look at step two for the rural sample. Indeed, with the FII data we find no significant effect, in either direction, of gender on ownership of an M-PESA account, but the FinAccess data indicate that female owners of a mobile phone are *more* likely to have an account. This is also true for the 'rural, vulnerable' sample. We come back to this below.

More generally, the results for step one indicate that a substantial part of the exclusion already happens at an early stage: the non-educated and the poor have a lower probability of owning a SIM/phone, across all samples. Where step two is concerned, apart from age, the key variable is education. For both datasets, education has a positive impact on the possession of an M-PESA account in the full sample (Figure 1 and Figure S1), but, interestingly, for the FII data—where we look at SIM owners—this effect comes from the rural subsamples (Figures 3 and 4), whereas for the FinAccess data—where we look at *phone* owners—it comes from the urban subsample (Figure S2). Finally, as already stressed in Section 6.3, in step three just about all variables now have a negative sign. This is also true for the FinAccess results, in spite of the different definition of saving (see Section 6.3).

## 7. Conclusions and Policy Implications

This paper has examined the uptake and use of M-PESA mobile financial services in Kenya. In particular, we wanted to find out to what extent and, more importantly, for which parts of the population the success of M-PESA as a money transfer mechanism has resulted in higher financial inclusion, which we equate with being able to save. To answer this question, we exploit survey data collected among 3000 respondents by InterMedia as part of the 2013 FII Program (and we also use FinAccess data in robustness checks). Specifically, we use a three-step probit procedure to identify the socio-demographic characteristics of, successively, the respondents who do not own a SIM card, own a SIM card but have not opened an M-PESA account, and, finally, have an M-PESA account but do not save on it.

Overall, the most important finding is that those who do not benefit from the positive effects of M-PESA (such as the ability to receive more frequent and faster remittances and, ultimately, the ability to save on a formal account) are disproportionally non-educated, poor, and female. The latter result is only indirectly visible in the FII data—in that women tend to have a lower education level—but comes to the forefront in the FinAccess data, which focus on mobile phone owners. With this dataset, we find that women are less likely to own a phone, and may thus already be excluded in the very first step. These findings put into perspective the results of, amongst others, Morawczynski and Pickens [31] and Suri and Jack [17] on the financial empowerment of women thanks to M-PESA. As mentioned in Section 3.3., Suri and Jack find that the beneficial effects of M-PESA are more pronounced for female-headed households, and they estimate that the spread of mobile money induced 185,000 women to switch into business or retail as their main occupation [17] (p. 1289). Our results do not contradict this but indicate that there are also women who are 'left behind'.

As such, our results go against the traditional optimistic discourse on the impact of mobile money on financial inclusion, as voiced by, for example, Ouma et al. [38] (p. 34; emphasis added): "deepening and expanding the scope of mobile financial savings is an avenue for promoting and mobilizing savings *particularly for the poor and low income earners* who have limited access to the formal banking system".

Our findings raise the questions why saving on MFS is not more prevalent in Kenya, what could be done to promote it, and, in particular, why M-PESA has failed to reach the 'poorest of the poor'. Obviously, whereas our analysis allows us to identify the socio-demographic characteristics that explain exclusion in each of the steps, we cannot always identify the precise nature of the underlying barriers. As Johnson and Arnold [28] (p. 720) point out (concerning step 2), "The ways

in which [socio-demographic] patterns of access are related to barriers to access are complex, as these may operate through combinations of discriminatory policies, informational and contractual frameworks, pricing and product features". For example, in a paper on Uganda, Murendo et al. [42] adopt a network-oriented explanation of MFS adoption, and argue that while social networks help spread information about MFS, the poorest households may reside in an "information-poor" situation, preventing them from adopting MFS. Our approach cannot, by its very nature, produce this type of insights.

However, it does bring other useful insights, with direct policy implications. In line with the philosophy behind our three-step, 'attrition' approach, let us list these insights step by step and let us then broaden the discussion by pointing out alternative/additional explanations proffered by other authors—in particular, with an eye on future research into the matter.

To start with step 1, let us first stress that where the rural and 'rural, vulnerable' samples are concerned, a substantial part of the exclusion already happens at this stage. As can be seen in Figure 3 and Figure S3, 22.7 per cent of the rural population did not own a SIM in 2013 and no less than 43.8 per cent did not own a phone. For the 'rural, vulnerable', these numbers are even higher—at 28.9 and 59.4 per cent, respectively; see Figure 4 and Figure S4. Second, our regressions in Table 2 and Table S2 show that those who do not own a SIM/phone are predominantly non-educated and poor, and (because women tend to be less educated) also female. Third, we also find that the AMEs of wealth are biggest for the 'rural, vulnerable' sample.

The first policy suggestion that we would like to put forward is therefore to give away mobile phones to the (rural) poor. This might sound odd in a country where the penetration of mobile subscriptions reached 94.3 per cent at end-2017 [57] (p. 7), but individuals can obviously 'multi-SIM' and, as Wyche and Olson [46] (p. 38) point out, nationwide statistics reveal little about spatial differences and fail to capture gendered inequalities in mobile phone ownership. A comparison of Figures S2 and S3/S4 clearly shows a rural/urban divide in phone ownership.

Free phones would not only help the poorest to overcome the first hurdle toward MFS-induced financial inclusion, it is also interesting to note that the attrition between step 1 and step 2 (ownership of an M-PESA account) is much lower for phone owners (in the FinAccess data in Figures S1–S4) than for the broader group of SIM owners (in the FII data in Figures 1–4), and this in particular in the rural samples. This suggests that phone ownership makes it easier to have one's own M-PESA account. Phone ownership may also attenuate the intra-household bargaining difficulties of women. Kusimba et al. [47] (p. 11) report on women who use their phone to hide money from their husbands. Last but not least, the results of a recent randomised control trial by Roessler et al. [58] among women from low-income households in Tanzania are particularly promising. One year into the experiment, women who had been given either basic handsets or smartphones were significantly more likely to use mobile money, use phones for income-generating activities, and score higher on an index of financial inclusion. Crucially, compliers in the phone treatment groups—that is, subjects who still owned a phone at endline—reported consumption increases of USD 12 to USD 20 per month, "which likely feels significant to the poor women in our study whose households were consuming $2.59 per day on average" [58] (p. 8). In view of the cost of the phones (USD 18 for the basic phones and USD 65 for the smartphones), Roessler et al. conclude that "the interventions produce a very high yield on investment and may well provide a cost-effective means of poverty reduction".

Turning to step 2, when looking for policy implications the positive impact of education catches the eye. (Age is also significant but is obviously not a policy variable.) In Figures 1–4 it can be seen that the higher the education level, the higher the probability that the owner of a SIM has an M-PESA account. Figures S1–S4—where we look at owners of a phone—paint a slightly different picture. For the samples that we are most interested in—'rural' and 'rural, vulnerable'—education is not significant overall (i.e., across all age categories) in step 2, but it is significant for specific categories (see Table S3). Education in all probability correlates with mobile phone skills. Moreover, in a paper on Uganda,

Kiconco, Rooks, Solana and Matzat [59] find that the possession of basic technical mobile phone skills has a strong effect on the odds of adopting mobile money.

There is also a paper—by Wyche, Simiyu and Otheno [60]—that examines the mobile phone skills of Kenyans, but, as we will explain, this paper might be more about step 3 than about step 2. Specifically, Wyche et al. rely on the amplification theory of technology—which holds that technology can amplify existing social inequalities—to study how rural Kenyan women interact with the products and services of Safaricom. Wyche et al. focus on women "because they tended to be less capable of using their phones (compared to men)" [60] (p. 6). Importantly, although it was not a criterion to be part of the sample, "all of the women were familiar with, and had used, M-PESA" [60] (p. 6). This does not necessarily imply that all of Wyche et al.'s respondents have an M-PESA account. As mentioned in Table 1, receiving money is possible without having an account. But quite a few women in the sample probably do have one. Hence our remark about the relevance of Wyche et al.'s study not being limited to step 2. Whatever the case, Wyche et al.'s interviews revealed a mismatch between the skills of rural female phone owners and, for example, the M-PESA interface. Worryingly, for some women even topping up a handset was overly complicated. As Wyche et al. [60] (p. 13) put it, "if women struggle just to add airtime to their handsets, it is unclear how they can take advantage of [more advanced] services"; that is, it is not very realistic to expect these women to save on MFS. Wyche et al. [60] conclude that their findings provide, not unlike the present paper, "a counter-narrative to the mostly optimistic stories about the possibilities of widespread mobile phone ownership throughout sub-Saharan Africa" (p. 15).

Wyche et al.'s findings indicate that better product design is needed to reach the non-educated poor. Wyche et al. note that "the mobile phones (*sic*) models we observed all had different interfaces, none of which were intuitive. [ . . . ] in all cases, users must [ . . . ] navigate hierarchical menus and then scroll to the bottom of a list of rarely-used applications [ . . . ] to find the Safaricom selection" [60] (p. 9). The findings of Kiconco et al. [59] on the importance of mobile phone skills suggest that education campaigns should also help. Note in this respect that in Batista and Vicente's [61] field experiment in rural Mozambique participating farmers were all given a mobile money 'information module', comprising a simple phone, a verbal introduction to the mKesh service, registration on mKesh by the enumerators, and seed money equivalent to USD 2 for a first, assisted deposit into the mKesh account.

Where the third and final step in our approach is concerned, we find in Section 6.3 that the proportion of savers is particularly low among the 'rural, vulnerable': only 4.2 per cent in the FII definition of saving (Figure 4); 14.7 per cent under the less strict FinAccess definition (Figure S4). Also, in the other samples, those who are better educated—and who should thus be in a better position to save on MFS—are *less* likely to do so, and this effect is stronger in the FinAccess data (where we consider only phone owners). This apparent reluctance to save on MFS is surprising in the sense that M-PESA lowers at least two barriers to saving, namely transaction costs and regulatory barriers [62]—the latter because of less strict 'know your customer' rules. In their study on Uganda, Munyegera and Matsumoto demonstrate that the mechanism behind the increase in saving as a result of mobile money use is effectively "a reduction in transaction costs—a combination of transportation fares and service charges" [43] (p. 45), in particular for rural households. Note in this respect that the respondents in our 'rural, vulnerable' subsample, by construction, do not have a bank account (unlike some in the other samples).

One obvious explanation for the low incidence of saving on MFS relates to the negative real interest rate on M-PESA accounts. Empirical evidence as to the impact of higher interest rates is mixed. A recent experiment in Kenya by Habyarimana and Jack [63] finds no systematic differences in savings behavior between the M-SHWARI condition and the condition where respondents had access to a lock savings account (LSA) with a 1 percentage point higher interest rate. (To be clear, this LSA was only added to the suite of M-PESA services in June 2014, and was thus not yet available at the time of collection of the data that we use.) The results of the Batista and Vicente [61] experiment in Mozambique mentioned earlier are more promising, in that the 'savings' treatment increased savings

in mKesh by 38–44 per cent in the first year. The savings treatment allowed individuals to receive a bonus of 20 per cent interest, paid in fertiliser, on the average balance on their mKesh account over a pre-specified period. Note that this is a substantially higher bonus than in the Habyarimana and Jack [63] experiment. But perhaps more importantly, Batista and Vicente also find evidence that the savings account may have enabled participating farmers to counteract social pressure to share resources.

In our view, this points to a crucial issue for future research: are the money transfers that result from the well-documented social links and obligations in developing economies negative or positive? If one takes a negative view, one can argue with Batista and Vicente [61] (p. 2; emphasis added) that mobile money may "by itself de-incentiviz[e] savings by facilitating transfers to other people, making them *more* vulnerable to social pressure within their social networks". This could then be an explanation for the low incidence of saving on MFS observed in step 3 above and would seem to indicate that access to a commitment device, as in the Batista and Vicente experiment, might be needed to guard savers from the pressure of social claimants. If, however, collective (semi-)informal saving and funds transfers through social networks are beneficial, mobile money operators (MNOs) could try to tailor their services to promote these—in line with a question raised by Karlan et al. [62] (p. 68), albeit in relation to banks rather than MNOs: "Can financial institutions capture/bottle (some) beneficial peer effects remotely, without imposing the substantial transaction costs involved in higher-touch approaches (that involve, for example, regular group meetings)?".

Looking at the literature, it becomes clear that (behavioural) economists mainly see social money transfers as a barrier to saving, in that social claimants can induce individuals to engage in concealment and/or save less [62] (p. 54). Anthropological research brings a different view—and, overall, strikes a more positive note. Kusimba et al.'s analysis of 12 family networks reveals that reciprocity in money transfer networks is important as a cultural practice: "Sending value is a way to save through others until the gift is returned, perhaps as a remittance or even in a different form" [47] (p. 7); see also Johnson and Krijtenburg [49,64]. This collective, and sometimes intergenerational, dimension of informal saving would appear to be under-researched in the mainstream literature.

To be clear: anthropological research is not blind to the potential negative dynamics of social obligations. As Kusimba et al. [65] (p. 275) observe, in cultures that value reciprocity there can effectively be pressure from family members and neighbours to recirculate funds: "A request cannot be denied; instead, one's phone must be shut off or 'lost' to avoid pressure for remittances. Older women see many visitors once word of a recent remittance spreads in the community; they often share their PINs and consequently avoid using their mobile wallets as a store of value". At the same time, research by Johnson and Krijtenburg into the practices of interpersonal transactions among low-income people in Kenya highlights their two-sided nature and "upliftment" dimension. For example, during their in-depth interviews respondents who had helped relatives with school fees reported that "they were now able to appeal to those they had helped for support with their own children's education, saying "now is the time to ... get the benefit" [64] (p. 581). Johnson and Krijtenburg also conduct a semantic analysis of the relevant vocabularies of the Kamba ethnic group. This again brings to the front the collective nature of informal saving. There proved to be no exact equivalent for the term 'save' in Kikamba, and informants used two distinct concepts of 'saving'. One revolves around something being "kept safe" and the second "refers to the way in which members of a group 'put together' (*kūmbanya*) their resources" [49] (p. 18); see also [64] (p. 582).

A recent initiative that aims to tap into this collective dimension of saving is Soma in Mozambique. Soma is a project of UX Information Technologies, a local start-up that focuses on *grupos de poupança*. According to UX, these savings groups can cost up to USD 70 per member to operate. UX aims to bring this down to as little as USD 8 by means of an app that records all sort of information on the groups and their members—"a trove of data that UX hopes will eventually connect the groups to a wider financial system" [66]. Let us end by pointing out the limitations of our research. Apart from the limitation of our methodology set out above, there are also limitations that stem from the nature of the data that we

use. For one, we have no information on informal savings and, thus, could not study whether M-PESA has triggered a transition from informal to formal saving. Second, our saving variables are rough: not only are they yes/no dummies (which, by definition, do not provide any indication as to the level of the savings), they also lack a time dimension. As explained, the survey only asked respondents whether they had 'ever' saved. Third, because of the absence of a precise time indication we could not analyse whether saving on M-PESA functions as a stepping stone toward saving on a bank account. However, in view of the limited number of respondents who save on M-PESA (6.8 per cent of the total sample) or, broader, on MFS (7.6 per cent), it is safe to state that the stepping-stone effect, if any, is only of marginal importance, especially so in rural areas. Indeed, whereas Kenyans are 'paying' massively with M-PESA (that is, transferring money), this is not yet true for saving—at least not in 2013. In other words, unlike bKash in Bangladesh, M-PESA has not yet—as one of the founders of bKash recently put it in *The Economist*—"become the collective mattress for all the common people" [67].

**Supplementary Materials:** The following are available online at http://www.mdpi.com/2071-1050/11/3/568/s1, Table S1: Descriptive statistics, Table S2: Possession of a Mobile Phone, FinAccess data, Table S3: Possession of Formal Accounts by Individuals Who Own a Mobile Phone, FinAccess data, Table S4: Saving on MFS—M-PESA Users, FinAccess data, Figure S1: Overview of Results, FinAccess—Urban + Rural, Figure S2: Overview of Results, FinAccess—Urban, Figure S3: Overview of Results, FinAccess—Rural, Figure S4: Overview of Results, FinAccess—Rural, Vulnerable.

**Author Contributions:** L.V.H.: conceptualisation, methodology, data analysis, writing; and A.D.: data curation, data analysis, visualisation.

**Funding:** This research received no external funding.

**Acknowledgments:** We are grateful to David Bounie, the Editor and three anonymous referees for helpful comments and suggestions, and to Abel François for his advice on the econometrics. We also thank InterMedia for providing us with the data. Data is available upon request from InterMedia (http://finclusion.org/data_fiinder/). Do files are available upon request from the authors.

**Conflicts of Interest:** The authors declare no conflict of interest.

## Appendix A

**Table A1.** Descriptive statistics, %.

|  | FII | FinAccess |
| --- | --- | --- |
| M-PESA users | 72.7 | 58.7 |
| Bank account owners | 28.2 | 27.1 |
| Age |  |  |
| 15–25 | 22.3 | 25.9 |
| 26–30 | 17.2 | 17.3 |
| 31–35 | 13.1 | 12.1 |
| 36–40 | 12.2 | 10.4 |
| 41–55 | 20.8 | 16.8 |
| Over 55 | 14.4 | 17.5 |
| Gender |  |  |
| Female | 62 | 59.1 |
| Male | 38 | 40.9 |
| Education |  |  |
| No education | 33.3 | 39.0 |
| Primary | 39.1 | 36.1 |
| Secondary | 25.6 | 22.8 |
| College | 2.0 | 2.1 |
| Urbanity (Urban = 1) | 36.7 | 35.9 |
| Employed | 70.4 | 79.8 |
| Wealth | 14.1 | 11.6 |
| N | 3000 | 6449 |

*Note*: In both surveys, 'Education' specifies whether the respondents have completed part or all of an education level. We classify them depending on the highest level *fully* completed. Regarding employment, the Financial Inclusion Insights (FII) survey asks: "Do you currently have a job that earns you an income? It does not matter if it is formal or informal, part-time or full-time." We classified as 'Employed', respondents who answered 'yes' to this question. The FinAccess survey asks about several types of occupations, and whether they are full-time, part time, or seasonal. We classified as 'Employed' all respondents answering 'yes' to at least one of these questions.

**Table A2.** Saving on a formal account.

| | Saving on MFS | | | | Saving on a Bank Account | | |
|---|---|---|---|---|---|---|---|
| | Urban + Rural (1) | Urban (2) | Rural (3) | Rural, Vulnerable (4) | Urban + Rural (5) | Urban (6) | Rural (7) |
| Outcome equation | | | | | | | |
| Age | n.s. | n.s. | n.s. | n.s. | n.s. | * | n.s. |
| 15–25 | – | – | – | – | – | – | – |
| 26–30 | 0.0982 | 0.0623 | 0.181 | 0.429 * | 0.437 ** | 0.604 *** | 0.0935 |
| | (0.92) | (0.45) | (1.05) | (2.06) | (3.19) | (3.62) | (0.35) |
| 31–35 | 0.0236 | −0.306 | 0.335 | 0.419 | 0.368 * | 0.285 | 0.552 * |
| | (0.20) | (−1.66) | (1.92) | (1.92) | (2.43) | (1.37) | (2.29) |
| 36–40 | 0.134 | 0.215 | 0.177 | 0.234 | 0.308 | 0.315 | 0.419 |
| | (1.11) | (1.22) | (1.00) | (1.00) | (1.94) | (1.38) | (1.71) |
| 41–55 | 0.00117 | 0.0726 | 0.0433 | 0.214 | 0.293 * | 0.323 | 0.373 |
| | (0.01) | (0.46) | (0.26) | (1.00) | (2.06) | (1.61) | (1.64) |
| Over 55 | −0.318 * | −0.137 | −0.292 | −0.271 | 0.366 * | 0.751 ** | 0.322 |
| | (−2.12) | (−0.58) | (−1.38) | (−0.83) | (2.28) | (3.21) | (1.27) |
| Gender (Female = 1) | −0.205 ** | −0.185 | −0.272 ** | −0.362 ** | −0.301 *** | −0.415 *** | −0.231 |
| | (−2.80) | (−1.71) | (−2.62) | (−2.65) | (−3.46) | (−3.40) | (−1.77) |
| Education | *** | * | *** | *** | *** | *** | *** |
| Non-educated | – | – | – | – | – | – | – |
| Primary | 0.475 *** | 0.371 * | 0.460 ** | 0.396 * | 0.322 * | 0.339 | 0.213 |
| | (4.36) | (2.03) | (3.21) | (2.27) | (2.43) | (1.48) | (1.20) |
| Secondary | 0.766 *** | 0.594 ** | 0.801 *** | 0.915 *** | 0.790 *** | 0.685 ** | 0.782 *** |
| | (6.68) | (3.15) | (5.27) | (4.66) | (5.84) | (2.98) | (4.31) |
| College | 0.271 | −0.118 | 0.588 | no variation | 1.316 *** | 1.131 *** | 1.304 *** |
| | (1.00) | (−0.31) | (1.44) | | (6.05) | (3.64) | (3.68) |
| Wealth | 0.0232 ** | 0.0104 | 0.0333 ** | 0.0403 ** | 0.0135 | 0.0183 | 0.0119 |
| | (3.26) | (1.00) | (3.26) | (3.02) | (1.61) | (1.47) | (0.98) |
| Family size | −0.0180 | −0.00335 | −0.00408 | 0.00393 | −0.0207 | −0.0342 | 0.0193 |
| | (−1.18) | (−0.13) | (−0.20) | (0.15) | (−1.14) | (−1.15) | (0.86) |
| Constant | −2.05 *** | −1.64 *** | −2.44 *** | −2.701 *** | −2.382 *** | −2.194 *** | −2.726 *** |
| | (−12.55) | (−6.64) | (−10.08) | (−8.83) | (−11.79) | (−7.01) | (−8.74) |
| Pseudo $R^2$ | 0.0768 | 0.0387 | 0.1102 | 0.1416 | 0.1053 | 0.1037 | 0.1073 |
| AIC | 1512.554 | 772.5237 | 736.0625 | 436.1237 | 1030.454 | 564.7305 | 455.7456 |
| BIC | 1584.607 | 832.5496 | 802.6262 | 493.7709 | 1102.506 | 624.7564 | 522.3093 |
| Log likelihood | −744.277 | −374.261 | −356.031 | −207.06187 | −503.2270 | −270.3652 | −215.8728 |
| Observations | 2994 | 1099 | 1895 | 1395 | 2994 | 1099 | 1895 |

*t* statistics in parentheses; *, ** and *** indicate significance at the 0.05, 0.01 and 0.001 level, respectively; asterisks next to the name of a categorical variable indicate its overall significance.

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
