# Peer review of "M-PESA and Financial Inclusion in Kenya: Of Paying Comes Saving?"

_sustainability, doi:10.3390/su11030568_

Reviewer 1 Report

This study examines M-PESA, a leading local mobile money transfer service, and financial inclusion in Kenya. The research agenda is interesting and could contribute to the literature on saving, financial inclusion, financial tools, mobile financial services.

Below, the main critical issues are presented:

 Introduction

The introduction is almost fine.  

I would use a different way to suggest the pages of the bibliographic reference, unless this is the usual mode of communication in the Journal guidelines.

 M-PESA identikit

Insert the acronym CBA after Commercial Bank of Africa in brackets so that it is possible to quote directly CBA afterwards.

 Data

As for Commercial Bank of Africa, it would be necessary to do the same thing for “Financial Inclusion Insights Program” and insert in brackets immediately after the initials FII.

 Methodology

Put in brackets on line 270 "see also [12]", same thing for line 289  "see also [12] (p.172)".

 Discussion

The images in figures 1, 2, 3 and 4 are not clearly seen, they are grainy.

 Conclusions

The conclusions are too long. There are too many comparisons that should be put in the discussions and not in the conclusions. In addition there are (for example in rows between 700 and 708) previous research cited but not discussed. Rearrange. They will have to be shorter and more punctual.

Author Response

Dear referee,

 Thank you very much for your comments. Please find our responses below.

For your convenience, in the manuscript we have indicated all revisions (including those made in response to other referees) in blue.  

 1. I would use a different way to suggest the pages of the bibliographic reference, unless this is the usual mode of communication in the Journal guidelines.

 Apparently this is the way the Journal does it. We have based ourselves on articles published earlier in Sustainability.

 2. Insert the acronym CBA after Commercial Bank of Africa in brackets so that it is possible to quote directly CBA afterwards.

 Done.

 3. […] it would be necessary to do the same thing for “Financial Inclusion Insights Program” and insert in brackets immediately after the initials FII.

 We already do that earlier; namely on line 56 (of the first version).

4. Put in brackets on line 270 "see also [12]", same thing for line 289 "see also [12] (p. 172)".

 Done. To be clear: in our version this was on lines 274 and 294.

 5. 
The images in figures 1, 2, 3 and 4 are not clearly seen, they are grainy.

 We assume that you have not seen the original Figures (which we uploaded separately from the manuscript), but only the Figures in the text. The latter are indeed grainy. This is because we have taken a screenshot of the original Figures and then pasted that screenshot in the text. The quality of the Figures in the published version – if our paper is accepted – should be much better.

 6. The conclusions are too long. There are too many comparisons that should be put in the discussions and not in the conclusions. In addition there are (for example in rows between 700 and 708) previous research cited but not discussed. Rearrange. They will have to be shorter and more punctual.

 Point taken. Now that we have had a fresh look at our paper we indeed see that the old Section 8 was too long and not well-structured. In response to your comment (and a similar comment by the Editor) we have

 - first, renamed and renumbered “7. Discussion” into “6.4. A View Across the Three Steps” and, by doing so, moved it to Section 6 (“Results”) – as this is where it really belongs.  Indeed, upon re-reading our paper we realised that Section 7 was not really a Discussion in the way it is usually done in academic articles; that is, a comparison of one’s own results with the results of others.

Rather, Section 7 (now 6.4) summarised (and discussed) the results presented in Sections 6.1-6.3;

 - completely changed the structure of the concluding Section. In order to make it more punctual, we now simply rely on our three-step approach to structure (the bulk of) our conclusions. That is, after a brief recap, we now discuss the results and the associated policy implications step by step; first for step 1, then for step 2, etc. In line with this new structure, we now also say more about relative importance of the different hurdles. After the step-by-step discussion, we “broaden the discussion by pointing out alternative/additional explanations proffered by other authors – in particular with an eye on future research into the matter”.

 And in order to make the conclusions shorter, we have substantially cut down the discussion of the studies by Roessler et al., Habyarimana and Jack, and Batista and Vicente, and we no longer mention Johnson and Johnson and Krijtenburg (at least not here).

 As for this specific remark of yours: “In addition there are (for example in rows between 700 and 708) previous research cited but not discussed”, we assume (given that we would appear to have different line numbers) that you refer to this paragraph (709-716 in our numbering):

 “Anthropological research brings a different view – and, overall, strikes a more positive note. As Kusimba, Yang, and Chawla [42] (p. 11) point out, social network analysis captures “dimensions that go beyond the individual perspective of questionnaire data and the urban sender/rural receiver dynamic of the ‘send money home’ view”. Their analysis of twelve family networks reveals that reciprocity in money transfer networks is important as a cultural practice: “Sending value is a way to save through others until the gift is returned, perhaps as a remittance or even in a different form” [42] (p. 7); see also Johnson and Krijtenburg [44] (p. 19). This collective, and sometimes intergenerational, dimension of informal saving would appear to be under-researched in the mainstream literature.”

It is correct that we do not explain just how they perform their social network analysis, but, in our view, this would lead us to far and it would break the flow of the text. We just wanted to highlight their main findings (reciprocity is important; there is a collective decision).

 Finally, we noted that you indicated that “Moderate English changes [are] required”. After consulting with the Journal, we will have our manuscript proofread by a native speaker if and when it is accepted.

Reviewer 2 Report

While it is an interesting paper describing the financial inclusion in Kenya in terms of local mobile money transfer, the contribution is unclear. The paper is rather descriptive than in-depth research. I'd suggest you to focus more on what implications and insights can be provided to researchers. The results are not surprising and they were discovered long ago in other countries. What new findings can you provide and in what perspective your study can push forward our knowledge? What factors indeed are preventing people moving towards the next step of financial inclusion? You described the facts but better to study the reasons behind.

Also you need to link financial inclusion to sustainability by reviewing the related literature. This may help clarify your contribution and position better in the literature.

Author Response

Dear referee,

 Thank you very much for your comments. Please find our responses below.

For your convenience, in the manuscript we have indicated all revisions (including those made in response to other referees) in blue

 1. …, the contribution is unclear. The paper is rather descriptive than in-depth research.

 We now explicitly list (what we think are) the contributions of the paper in the Introduction (lines 75-80):

 “Our paper contributes to the literature in several respects. In terms of data, we are the first to use the FII dataset – at least for our purposes. In terms of methodology, compared to the extant studies – which examine only one of the three steps and/or lump together some or all of the steps – our three-step approach allows us to map the relative importance of the different hurdles and identify more accurately the determinants of the ‘attrition’ at each of the stages. In terms of findings, we bring a sobering note to the more optimistic results of, in particular, Suri and Jack [17], …”

 The novelty and benefits of our three-step approach is something that we underpin at different points in the paper:

 (289-296) “To our knowledge, this is a novel approach. In the literature, the dominant method consists in using dummies – for both steps two and three of our approach (see also [23]). Johnson and Arnold [28] (p. 741-742) have in their regressions for M-PESA adoption – i.e., step two of our approach – dummies for owning and having access to a phone. Kikulwe et al. [33] (p. 7) and Munyegera and Matsumoto [42] (p. 130) – in a paper on Uganda – also use a dummy for mobile phone ownership. Murendo, Wollni, de Brauw, and Mugabi [43], in another paper on Uganda, use the number of mobile phones owned by the household. We will show that the use of dummies for mobile phone ownership hides a number of interesting relationships between variables.”

(455-460) “From a methodological angle, let us stress that, in their paper on Uganda, Murendo et al. [43] also find that poor households are less likely to own a phone, but only indirectly. They find that the number of mobile phones owned has a significant positive impact on the adoption of mobile money, and when they split their sample into poor and non-poor households it becomes clear that the impact is bigger for poor households [43] (p. 338, Table 5). Our three-step approach has the advantage of directly uncovering such relationships.”

 (605-612) “The goal here is primarily to highlight the value-added of our three-step approach. Indeed, as we illustrate below, because an explanatory variable can be crucial in one step but not in the next or, stronger, because the direction of its impact can differ between steps, there may be little or no trace of its influence in a direct estimation as in Table A2. For example, for both the FII and FinAccess data, age consistently has a significant positive impact in (one of) the first two steps but a negative impact in step three, so that its effect is absent in the direct analysis. The results for gender also differ between the step-wise and the direct approach.”

 (613-627) “In terms of methodology, the latter observation may help understand why Johnson et al. [57] and Johnson and Arnold [28] find different results for gender. … In our view, this is another illustration of the value added of our three-step approach.”

 2. I'd suggest you to focus more on what implications and insights can be provided to researchers. … What factors indeed are preventing people moving towards the next step of financial inclusion? You described the facts but better to study the reasons behind.

 We hope that the new structure of the concluding section remedies this. In order to give the section more ‘direction’, we now simply use our three-step approach as the basic framework. That is, after a brief recap, we now discuss the results and the associated policy implications step by step; first for step 1, then for step 2, etc.

 In line with the spirit of this new structure, we now also say more about the relative importance of the different hurdles:

 “To start with step 1, let us first stress that where the rural and ‘rural, vulnerable’ samples are concerned, a substantial part of the exclusion already happens at this stage. As can be seen in Figures 3 and S3, 22.7 per cent of the rural population did not own a SIM in 2013 and no less than 43.8 per cent did not own a phone. For the ‘rural, vulnerable’, these numbers are even higher – at 28.9 and 59.4 per cent, respectively; see Figures 4 and S4.”

 “Where the third and final step in our approach is concerned, we find, in Section 6.3, that the proportion of savers is particularly low among the ‘rural, vulnerable’: only 4.2 per cent in the FII definition of saving (Figure 4); 14.7 per cent under the less strict FinAccess definition (Figure S4).”

 After the step-by-step discussion, we “broaden the discussion by pointing out alternative/additional explanations proffered by other authors – in particular with an eye on future research into the matter” (emphasis added).

 We hope that this new approach makes clearer the links between our results and the policy suggestions. In our mind these connections were clear (and in the text we referred here and there to this or that step), but given that both you and the Editor did not see the links, we clearly did not do a good enough job.

 3. The results are not surprising and they were discovered long ago in other countries.

 Yes and no. Some of our results are indeed in line with earlier results. However, for one, to the best of our knowledge, no other paper analyses all three steps.  Our approach has the benefit of showing which socio-demographic characteristics play a role in what step. Other papers sometimes lump together two (or even worse: all three) steps and, as a result, their findings are less precise (example 1) or even incorrect (example 2):

 (297-301) “Where step three of our approach is concerned, Munyegera and Matsumoto [44] (p. 51) have in their probit regression for the saving behaviour of Ugandan households a dummy variable that takes a value of 1 if the household has at least one member who use mobile money services. They do not initially control for ownership of a SIM of a mobile phone – so that the non-use of mobile money services can have very diverse causes – …

 (297-301) “Finally, there are also authors who simply ignore the preconditions: in their regressions for the saving behaviour of Kenyan households, Ouma et al. [38] do not control for ownership of a mobile phone or a SIM card”.

 Second, as Aron (2018, p. 167) points out, “both the setting and the characteristics of the MFS schemes may differ across countries”. One can therefore not assume that what holds for one country will also hold for another.

 Third, given the explosive growth of M-PESA, it makes sense to have a second look at earlier results for Kenya to see whether they still hold; that is, to see whether the causes of the financial exclusion are still the same.

 Fourth, while the urban/rural split has been made before, we do not think anyone else has used a ‘rural vulnerable’ sample yet.

 Finally, related to the previous point, some of our results are surprising.  This is an example:

 (440-442) A less intuitive difference is that both gender and family size lose their significance. In particular, one would expect access to a phone to be more of problem for rural than for urban women. We come back to this below.

 4. Also you need to link financial inclusion to sustainability by reviewing the related literature. This may help clarify your contribution and position better in the literature.

 The Editor made a similar remark. In the Introduction we now explicitly link financial inclusion with sustainability, as well as with the UN’s Sustainable Development Goals (lines 36-39).

 We have, however, refrained from “reviewing the related literature” as we thought this would lead us to far. We do not really have room for such an overview, given that our paper is already quite long.

 Finally, concerning your remark that “Extensive editing of English language and style [is] required
”, we have been in touch with the Journal and we will have our manuscript proofread by a native speaker if and when it is accepted.

Reviewer 3 Report

The manuscript considers the adoption of saving behaviors in Kenya through the access to the M-PESA, among a set of Mobile Financial Services (MFS). The observational evidence suggests that just a small number of potential savers, that is, individuals who have access to MFS, are actually saving. The analysis focuses on socio-economic individual characteristics and it does well, with the support of a wide range of arguments on financial tools, economic development and even ethnography. Data from a survey are used in a three-step empirical exercise: first, access to SIM card; second, access to the M-PESA; third, saving practices. The sample is also subset to focus on the most critical groups: rural and vulnerable population. Possible selection biases are carefully taken into consideration setting up a Heckman selection model. Robustness checks are included in appendix based on a different collection of data. Results support to conclude that self-selection is strong for the M-PESA to generate a positive impact (non-educated, poor and female are penalized). These results are in line with some of the evidence in literature but more clear in some way. The authors claim and document indeed that the empirical settings are one of the most original contribution to the literature that allows to very stress evidence.

The topic proposed is interesting and materials are treated properly. Techniques are motivated and set up with care. I think nonetheless that the overall presentation is not yet at the level with contents. I fell the presentation as sometimes redundant. Appropriate changes in the structure of the manuscript may help a more synthetic and linear presentation. Especially for sections 2 to 4. This is crucial in my opinion, because the manuscript is very reach of arguments and materials. Also the English phrasing sounds sometimes unusual. Professional proofreading is recommended. In my opinion, a number of minor revisions are therefore in order. My comments are as follows (line numbers or document parts at the begging of each comment).

Title – The use of "of" sounds inappropriate. Maybe better "from"? See also line 92.

23 – As financial inclusion is a core element of the background, the definition could be supported even now by discussion and reference to the literature. One definition appears the first time at lines 87–88.

26 – Formal, semi-formal, and informal services/sectors could be better defined.

29 – The use of "while" sounds inappropriate. Maybe better "despite"? In addition, where a word is stressed as with partial italics in this case, more insights into the topic are expected.

32 – One additional line could motivate why it would.

39 – Some data could support readers to understand the topic.

42 – This information is already reported at line 26. So, I would start from introducing the supply side with "As mentioned, ..." before the demand side.

56 to 68 – I wouldn't anticipate results and conclusions so extensively here.

57 – References are in order.

84 – The acronym CBA should be introduced earlier at line 83.

94 – CPMI requires spelling.

Section 3 – I would prefer a literature review on the adoption of the MFS (Section 3 in this version) before the more specific information on the selected case for empirical exercise (Section 2 in this version). Also a summary table could help systematize the main insights into the topic.

113 – Readers who are not very used to the topic may want to know why and in which extent the MFS may differ across countries. Furthermore, they may want to know why Kenya is a case of relevance. I would not postpone this discussion.

Section 4 – This section could be appended to current Section 2.

246 to 254 – I would move these lines to the Appendix. The same is for lines 521 to 528 and 580 to 590. In general, I would do it each time comments refer to the materials in the Appendix. It is difficult indeed to follow comments to materials in the Appendix. I therefore suggest to remove all the comments of this sort from the main text (and present them in the Appendix).

501 – I am not surprised as it can be expected based on the empirical settings. 

Section 7 – Several elements for a discussion are presented together with results in current Section 6. I would suggest to merge the two into one "Results and Discussion". 

Section 8 – I would prefer that limitations are presented before policy implications. The reason is that limitations can serve to size the strength and reliability of recommendations delivered. I would also suggest more synthesis for the overall section.

Author Response

Dear referee,

 Thank you very much for your comments. Please find our responses below.

For your convenience, in the manuscript we have indicated all revisions (including those made in response to other referees) in blue

 1. Title – The use of "of" sounds inappropriate. Maybe better "from"? See also line 92.

 We are aware that it may sound odd at first, but that is because – as we indicate in line 92 – we paraphrase an old proverb. If we change “of” into “from”, we lose the link with the proverb. So, unless the Editor has strong feelings about it, we would prefer not to tinker with the title.

 23 – As financial inclusion is a core element of the background, the definition could be supported even now by discussion and reference to the literature. One definition appears the first time at lines 87–88.

 We see your point. However, we think that starting a paper with a conceptual discussion does not make for an attractive intro. In our view, the general description of ‘financial inclusion’ that we had in the first sentences was sufficient to set the scene in an Introduction and gave readers a good enough understanding until they arrived at the more detailed definition in Section 2.

 As an intermediate solution, we now use Klapper and Singer’s definition in the first sentence but leave the discussion for Section 2 – in order not to break the flow of the introduction.

 26 – Formal, semi-formal, and informal services/sectors could be better defined.

 We have added examples between brackets and refer the reader to [2] for more precise definitions.

 29 – The use of "while" sounds inappropriate. Maybe better "despite"? In addition, where a word is stressed as with partial italics in this case, more insights into the topic are expected.

 We have changed “While” into “Although”. The partial italics on “informal” were only meant to better contrast it with “formal” in the second part of the sentence. The italics have been removed (as they were not really necessary).

 32 – One additional line could motivate why it would.

 We agree. In fact, that line was there in an earlier version but was axed when trying to make the paper shorter. We have re-introduced the sentence and added a second one – because Reviewer 2 asked us “to link financial inclusion to sustainability” and the Editor wanted “to see a better framing of the paper
 within the sustainability literature”.  In view of this, we now also explicitly link financial inclusion with the 
UN’s sustainable development goals.

 39 – Some data could support readers to understand the topic.

 Point taken. We now cite World Bank figures on poverty in Kenya and FAO statistics on the importance of agriculture.

 42 – This information is already reported at line 26. So, I would start from introducing the supply side with "As mentioned, ..." before the demand side.

 Done.

 56 to 68 – I wouldn't anticipate results and conclusions so extensively here.

 We can see your point, but we received mixed signals here. Both the Editor and especially Reviewer 2 wanted us to stress the contribution of our paper.  And this proved difficult without already mentioning our findings (as we wanted to highlight that our “results go against the traditional optimistic discourse on mobile savings as a prime path to financial inclusion”).

 57 – References are in order.

 Done.

 84 – The acronym CBA should be introduced earlier at line 83.

 Done.

 94 – CPMI requires spelling.

 Done.

 Section 3 – I would prefer a literature review on the adoption of the MFS (Section 3 in this version) before the more specific information on the selected case for empirical exercise (Section 2 in this version). Also a summary table could help systematize the main insights into the topic.

 Now that you have pointed it out, we can see that, judging from the titles of the Sections, the sequence “M-PESA Identikit” (Section 2) – “Related Literature” (Section 3) – “Data” (Section 4) might come across as not being sufficiently “linear”, as you call it.

 However, as we stress in the introduction to the section, in Section 3 “we focus exclusively on Kenya”. So it is not as if we go from the specific to the more general and back again.  Also, knowledge of what M-PESA is, and how it has evolved over time, is needed for a good understanding of the results in the literature. It helps understand, for example, why the results of early studies differ from those of more recent ones. We have therefore left the order of Sections 2-4 unchanged. We have, however, changed the title of Section 3 into “State of the literature on M-PESA” – in order not to create any false expectations among readers. For the same reason, when we explain the structure of the paper in the final paragraph of the Introduction, we now also explicitly say that Section 3 focuses on Kenya.

 113 – Readers who are not very used to the topic may want to know why and in which extent the MFS may differ across countries. Furthermore, they may want to know why Kenya is a case of relevance. I would not postpone this discussion.

 We agree that a geographically broader literature review/description could, in principle, add value. However, we felt (and continue to feel) that we simply did not have the room for such an overview, as our paper is already quite long. In this respect, please also note that both the Editor and Reviewers 1 and 2 asked us to cut down on the "literature review-like material” in Section 8.However, in view of your remark that readers “may want to know why Kenya is a case of relevance” we have added a sentence to the Introduction to highlight that M-PESA is often presented as the poster child of financial inclusion thanks to mobile money adoption, both in the financial press and in the academic literature. Please see lines 100-102.

Section 4 – This section could be appended to current Section 2.

 Please see above.

 246 to 254 – I would move these lines to the Appendix. The same is for lines 521 to 528 and 580 to 590. In general, I would do it each time comments refer to the materials in the Appendix. It is difficult indeed to follow comments to materials in the Appendix. I therefore suggest to remove all the comments of this sort from the main text (and present them in the Appendix).

 We have given this serious thought and we even started implementing it. However, while it was manageable for the material in the Appendix, it did not work well for the discussion of Tables S2, S3 etc. in the Supplementary Materials. As the latter will only be available separately on-line, chances are that if we were to put the discussion in the Supplementary Materials, many readers might not consult it – or at least not while reading the paper for the first time. This would render problematic any later reference to results obtained with the FinAccess dataset. Also, the FinAccess dataset serves as a robustness check for the results obtained with the FII data, which means that we have to compare results, and, in our view, this is best done in one spot – rather than split over the main text and the Supplementary Materials.

 In our view, readers should be able to follow the discussion of Tables S2, S3 etc. without actually consulting the Tables. (We have removed one occurrence of “As can be seen” (in line 266 in the new version), in order not to create the impression that readers should consult the table.)

 501 – I am not surprised as it can be expected based on the empirical settings.

 We assume that you refer to this sentence: “Perhaps somewhat surprisingly, while ‘wealth’ is positively correlated to adoption it is only significant in the full sample” (line 506 in our version). As we indicate in the sentences that follow, it is indeed logical but perhaps not all readers see that immediately.

 Section 7 – Several elements for a discussion are presented together with results in current Section 6. I would suggest to merge the two into one "Results and Discussion".

 Point taken. In reply to this comment and similar comments by the Editor and Reviewer 1, we have

 - first, renamed and renumbered “7. Discussion” into “6.4. A View Across the Three Steps” and, by doing so, moved it to Section 6 (“Results”) – as this is where it really belongs. Indeed, upon re-reading our paper we realised that Section 7 was not really a Discussion in the way it is usually done in academic articles; that is, a comparison of one’s own results with the results of others.  Rather, Section 7 (now 6.4) summarised (and discussed) the results presented in Sections 6.1-6.3;

 - completely changed the structure of the concluding Section. In order to make it more punctual, we now simply rely on our three-step approach to structure (the bulk of) our conclusions. That is, after a brief recap, we now discuss the results and the associated policy implications step by step; first for step 1, then for step 2, etc. In line with this new structure, we now also say more about the relative importance of the different hurdles. After the step-by-step discussion, we “broaden the discussion by pointing out alternative/additional explanations proffered by other authors – in particular with an eye on future research into the matter”.

 Section 8 – I would prefer that limitations are presented before policy implications. The reason is that limitations can serve to size the strength and reliability of recommendations delivered.

 In response to this comment, we have signal the arguably most important limitation before we bring our policy suggestions.  Please see lines

 “Obviously, whereas our analysis allows us to identify the socio-demographic characteristics that explain exclusion in each of the steps, we cannot always identify the precise nature of the underlying barriers. As Johnson and Arnold [28] (p. 720) point out …  Our approach cannot, by its very nature, produce this type of insights.

However, it does bring other useful insights, with direct policy implications. …”

 The remaining limitations are still at the end of the paper, as this worked best in our view.

 Section 8 – I would also suggest more synthesis for the overall section.

 We have substantially cut down the discussion of the studies by Roessler et al., Habyarimana and Jack, and Batista and Vicente, and we no longer mention Johnson and Johnson and Krijtenburg (at least not here).

 Finally, concerning your remark that “Professional proofreading is recommended”, we have been in touch with the Journal and we will have our manuscript proofread by a native speaker if and when it is accepted.